# META DISCOVERY: LEARNING TO DISCOVER NOVEL CLASSES GIVEN VERY LIMITED DATA

**Haoang Chi**$^{1,2*}$ **Feng Liu**$^{3*}$  **Bo Han**$^{2}$  **Wenjing Yang**$^{1\dagger}$ **Long Lan**$^{1\dagger}$ **Tongliang Liu**$^{4}$
**Gang Niu**$^{5}$  **Mingyuan Zhou**$^{6}$  **Masashi Sugiyama**$^{5,7}$
$^1$National University of Defense Technology  $^2$Hong Kong Baptist University
$^3$University of Technology Sydney  $^4$The University of Sydney  $^5$RIKEN AIP
$^6$The University of Texas at Austin  $^7$The University of Tokyo
{haoangchi618,fengliu.ml,gang.niu.ml}@gmail.com, bhanml@comp.hkbu.edu.hk,
{wenjing.yang,long.lan}@nudt.edu.cn, tongliang.liu@sydney.edu.au,
mingyuan.zhou@mccombs.utexas.edu, sugi@k.u-tokyo.ac.jp

## ABSTRACT

In *novel class discovery* (NCD), we are given *labeled* data from *seen* classes and *unlabeled* data from *unseen* classes, and we train clustering models for the unseen classes. However, the implicit assumptions behind NCD are still *unclear*. In this paper, we demystify assumptions behind NCD and find that high-level semantic features should be shared among the seen and unseen classes. Based on this finding, NCD is theoretically solvable under certain assumptions and can be naturally linked to *meta-learning* that has exactly the same assumption as NCD. Thus, we can empirically solve the NCD problem by meta-learning algorithms after slight modifications. This meta-learning-based methodology significantly reduces the *amount of unlabeled data* needed for training and makes it more practical, as demonstrated in experiments. The use of *very limited data* is also justified by the application scenario of NCD: since it is unnatural to label only seen-class data, NCD is *sampling* instead of *labeling* in causality. Therefore, unseen-class data should be collected on the way of collecting seen-class data, which is why they are novel and first need to be clustered.

## 1 INTRODUCTION

With the development of high-performance computing, we can train deep networks to achieve various tasks well (Deng et al., 2009; Liu & Tao, 2015; Song et al., 2019; Jing et al., 2020; Song et al., 2020; Liu et al., 2020b; Han et al., 2020a; Xia et al., 2020). However, the trained networks can only recognize the classes seen in the training set (i.e., known/seen classes), and cannot identify and cluster novel classes (i.e., unseen classes) like human beings. A prime example is that human can easily tell a novel animal category (e.g., okapi) after learning a few seen animal categories (e.g., horse and dog). Namely, human can effortlessly *discover (cluster) novel categories* of animals. Inspired by this fact, previous works formulated a novel problem called *novel class discovery* (NCD) (Hsu et al., 2018; Han et al., 2019), where we train a clustering model using *plenty of* unlabeled novel-class and labeled known-class data.

However, if NCD is labeling in causality ($X \rightarrow Y$), there exists two issues: NCD might not be a theoretically solvable problem. For example, if novel classes are completely different from known classes, then it is unrealistic to use the known classes (like animals) to help precisely cluster novel classes (like cars, Figure 1(a)). Moreover, NCD might not be a realistic problem in some scenarios where novel classes might only be seen once or twice. This does not satisfy the assumptions considered in existing NCD methods. These issues naturally motivate us to find out when NCD can be theoretically solved and what assumptions are considered behind NCD.

In this paper, we revisit NCD and find that NCD will be a well-defined problem if NCD is *sampling in causality* ($Y \rightarrow X$), i.e., novel and known classes are sampled together (Figure 1(b)). In this

---

$^*$Equal contribution. Work done when Haoang Chi remotely visited HKBU.
$^\dagger$Corresponding author.

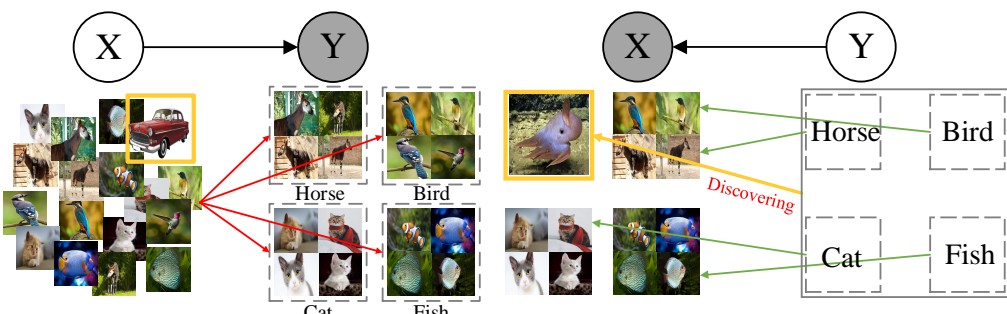

(a) Experts annotate data (labelling in causality).     (b) Experts sample data (sampling in causality).

Figure 1: NCD aims to discover novel classes (i.e., clustering novel-class data) with the help of labeled known-class data. There exists two ways to obtain data in NCD: (a) labeling in causality, e.g., we first obtain unlabeled images and then hire experts to label them and (b) sampling in causality, e.g., we are given a label set, and then sample images regarding these labels. In (a), experts have to go through all images and find out novel classes. However, the novel classes (like cars) might be totally different from known classes (like animals), which makes NCD become a theoretically unsolvable problem. In this paper, we revisit NCD from (b), where novel-class data are collected on the same way of sampling known-class data. In this view, NCD can be theoretically solved, since novel classes and known classes are highly related. The yellow rectangles represent the identified *novel* classes.

sampling process, data are often obtained because of a given purpose, and novel classes and known classes are obtained in the same scenario. For instance, botanists sample plant specimens for research purposes in the forests. Except for the plants they are interested in (i.e., known classes), they also find scarce plants never seen before (i.e., finding novel classes). Since a trip to forests is relatively costly and toilsome, botanists had better sampled these scarce plants passingly for future research. From this example, it can be seen that botanists will have plenty of labeled data with known classes, and few unlabeled data with novel classes. Since both data are sampled together from the same scenario (e.g., plants in the forests), it is reasonable to leverage knowledge of known classes to help cluster novel classes, which is like "discovering novel categories" happened in our daily life.

Therefore, we argue that the key assumption behind NCD is that, known classes and novel classes *share high-level semantic features*. For example, known classes and novel classes are different plants but all of them have the leaf, the stem, and the roots. Then, we reformulate the NCD problem and show that NCD is theoretically solvable under mild assumptions (including the key assumption above). We also show an impossibility theorem for previous NCD setting. This theorem shows that it might not be necessary to introduce known-class data to help cluster novel-class data if known classes and novel classes do *not* share high-level semantic features. Namely, NCD might be an ill-defined problem if known and novel class do *not* share high-level semantic features.

Although NCD is theoretically solvable under mild assumptions, we still need abundant data from known and novel classes to ensure NCD can be empirically solved. However, as we mentioned previously, the novel classes might only be seen once or twice in some scenarios. In such scenarios, we find that previous NCD methods do not work well (Figure 2). To address the *NCD given very limited data* (NCDL), we link NCD to meta-learning that also assumes that known and unknown (novel) classes share the high-level semantic features (Maurer, 2005; Chen et al., 2020a).

The key difference between meta-learning and NCDL lies in their inner-tasks. In meta-learning, the inner-task is a classification task, while in NCDL, it is a clustering task. Thus, we can modify the training strategies of the inner-tasks of meta-learning methods such that they can discover novel classes, i.e., *meta discovery*. Specifically, we first propose a novel method to sample training tasks for meta-learning methods. In sampled tasks, labeled and unlabeled data share high-level semantic features and the same clustering rule (Figure 3). Then, based on this novel sampling method, we realize meta discovery using two representative meta-learning methods: *model-agnostic meta-learning* (MAML) (Finn et al., 2017) and *prototypical network* (ProtoNet) (Snell et al., 2017). Figure 2 demonstrates two realizations of meta discovery (i.e., *meta discovery with MAML* (MM) and *meta discovery with ProtoNet* (MP)) can perform much better than existing methods in NCDL.

We conduct experiments on four benchmarks and compare our method with five competitive baselines (MacQueen et al., 1967; Hsu et al., 2018; 2019; Han et al., 2019; 2020b). Empirical results show that our method outperforms these baselines significantly when novel-class data are very limited.

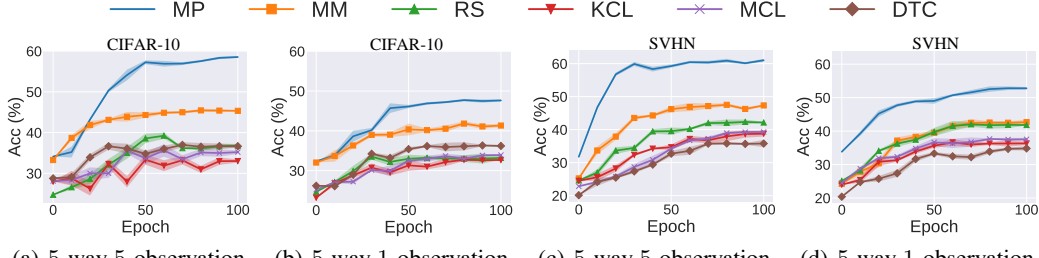

| (a) 5-way 5-observation | (b) 5-way 1-observation | (c) 5-way 5-observation | (d) 5-way 1-observation |

Figure 2: We conducted experiments on CIFAR-10 and SVHN and reported the *average clustering accuracy* (ACC (%), Section 5) when using existing and our methods to address the NCDL. We can observe that existing methods cannot address the NCDL, while our methods (MM and MP) can address the NCDL well.

Moreover, we provide a new practical application to the prosperous meta-learning community (Chen et al., 2020a), which lights up a novel road for NCDL.

## 2 RELATED WORK

Our proposal is mainly related to NCD, transfer learning and meta-learning that are briefly reviewed here. More detailed reviews can be found in Appendix A.

**Novel class discovery.** NCD was proposed in recent years, aiming to cluster unlabeled novel-class data according to underlying categories. Compared to unsupervised learning (Barlow, 1989), NCD also requires labeled known-class data to help cluster novel-class data. The pioneering methods include the *KL-Divergence-based contrastive loss* (KCL) (Hsu et al., 2018), the *meta classification likelihood* (MCL) (Hsu et al., 2019), *deep transfer clustering* (DTC) (Han et al., 2019), and the *rank statistics* (RS) (Han et al., 2020b). Compared to existing works (Hsu et al., 2018; Zhong et al., 2021c;b), we focus on clustering unlabeled data when their quantity is very limited.

**Transfer learning.** Transfer learning aims to leverage knowledge contained in source domains to improve the performance of tasks in a target domain, where both domains are similar but different (Gong et al., 2016). Representative transfer learning works are domain adaptation (Long et al., 2018) and hypothesis transfer (Liang et al., 2020), where mainly focus on classification or prediction tasks in the target domain. NCD problem can be also regarded a transfer learning problem that aims to complete the clustering task in a target domain via leveraging knowledge in source domains.

**Meta-learning.** Meta-learning is also known as learning-to-learn, which trains a meta-model over a wide variety of learning tasks (Ravi & Larochelle, 2017; Chen et al., 2020b; Yao et al., 2021; Wei et al., 2021). In meta-learning, we often assume that data share the same high-level features, which ensures that meta-learning can be theoretically addressed (Maurer, 2005). According to Hospedales et al. (2020), there are three common approaches to meta-learning: optimization-based (Finn et al., 2017), model-based (Santoro et al., 2016), and metric-based (Snell et al., 2017). In Jiang & Verma (2019), researchers trained a recurrent model that learns how to cluster given multiple types of training datasets. Compared to our meta discovery, Jiang & Verma (2019) learn a clustering model with multiple unlabeled datasets, while meta discovery aims to learn a clustering model with labeled known data and (abundant or few) unlabeled novel data from the same dataset.

## 3 ASSUMPTIONS BEHIND NCD AND ANALYSIS OF SOLVABILITY

Since NCD focuses on clustering novel-class data, we first show definitions regarding the separation of a *random variable* (r.v.) $X \sim \mathbb{P}_X$ defined on $\mathcal{X} \subset \mathbb{R}^d$. Then, we give a formal definition of NCD and introduce assumptions behind NCD. Finally, we present one theorem to show NCD is solvable in theory and one theorem to show a failure situation where previous setting encounters. The proofs of both theorems can be seen in Appendix B.

**Definition 1** ($K$-$\epsilon$-Separable r.v.). *Given the r.v. $X \sim \mathbb{P}_X$, $X$ is $K$-$\epsilon$-separable with a non-empty function set $\mathcal{F} = \{f : \mathcal{X} \to \mathcal{I}\}$ if $\forall f \in \mathcal{F}$*

$$\tau(X, f(X)) := \max_{i,j \in \mathcal{I}, i \neq j} \mathbb{P}_X(R_{X|f(X)=i} \cap R_{X|f(X)=j}) = \epsilon, \tag{1}$$

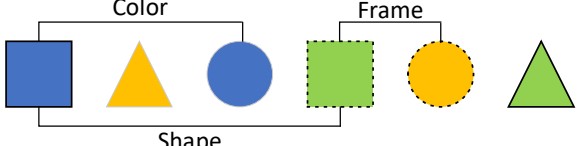

We can cluster these objects using at least three rules, i.e., colors, shapes and frames. Motivated by the rule of clustering, when addressing the NCDL problem, we need to sample inner-tasks, where data share the same rule (Algorithm 1).

Figure 3: When sampling tasks for meta discovery, we need to care about clustering rules.

*where $\mathcal{I} = \{i_1, \dots, i_K\}$ is an index set, $f(X)$ is an induced r.v. whose source of randomness is $X$ exclusively, and $R_{X|f(X)=i} = supp(\mathbb{P}_{X|f(X)=i})$ is the support set of $\mathbb{P}_{X|f(X)=i}$.*

In equation 1, $\tau(\cdot,\cdot)$ represents the largest overlap between any different clusters in the sense of the probability measure $\mathbb{P}_X$. If $\epsilon = 0$, then we can perfectly partition observations of $X$ into $K$ clusters using some distance-based clustering algorithm, e.g., K-means (MacQueen et al., 1967). However, when $\mathcal{X}$ is a complex space and the dimension $d$ is much larger than the intrinsic dimension of $X$ (e.g., images), it is not reliable to measure the distance between observations from $X$ using original features (Liu et al., 2020b), which will result in poor clustering performance.

**Non-linear transformation.** To overcome issues caused by the complex space, researchers suggest apply a *non-linear transformation* to extract high-level features of $X$ (Fang et al., 2020; Liu et al., 2020b). Based on these features, we can measure the distance between two observations well (Liu et al., 2020b). Let $\pi : \mathcal{X} \to \mathbb{R}^{d_r}$ be a transformation where $d_r$ is the reduced dimension and $d_r \ll d$, and we expect that the transformed r.v. $\pi(X)$ can be $K$-$\epsilon$-separable as well. Namely, we hope the following function set exists.

**Definition 2** (Consistent $K$-$\epsilon$-separable Transformation Set). *Given r.v. $X$ that is $K$-$\epsilon$-separable with a function set $\mathcal{F}$, a transformed r.v. $\pi(X)$ is $K$-$\epsilon$-separable with $\mathcal{F}$ if $\forall f \in \mathcal{F}$,*

$$\tau(\pi(X), f(X)) := \max_{i,j \in \mathcal{I}, i \neq j} \mathbb{P}_{\pi(X)}(R_{\pi(X)|f(X)=i} \cap R_{\pi(X)|f(X)=j}) = \epsilon, \tag{2}$$

*where $\pi : \mathcal{X} \to \mathbb{R}^{d_r}$ is a transformation. Then, a consistent $K$-$\epsilon$-separable transformation set is a non-empty set $\Pi$ satisfying that $\forall f \in \mathcal{F}, \forall \pi \in \Pi, \tau(\pi(X), f(X)) = \epsilon$.*

**Remark 1.** If $d_r \ll d$ and $\epsilon = 0$, we will need much less observations to estimate the density of $\pi(X)$ compared to $X$, thus it will be much easier to perfectly separate $\pi(X)$ than $X$. For example, there probably exists a linear function $g : \mathbb{R}^{d_r} \to \mathcal{I}$ such that $f(X) = g \circ \pi(X)$. It is clear that such linear function $g$ is easier to find than directly finding $f$ using K-means.

**Problem Setup of NCD.** Based on the above definitions, we will formally define the NCD problem below. In NCD, we have two r.v.s $X^l$, $X^u$ defined on $\mathcal{X}$, the ground-truth labeling function $f^l : \mathcal{X} \to \mathcal{Y}$ for $X^l$ and a function set $\mathcal{F} = \{f : \mathcal{X} \to \mathcal{I}\}$, where $\mathcal{Y} = \{i_1^l, \dots, i_{K^l}^l\}$ and $\mathcal{I} = \{i_1^u, \dots, i_{K^u}^u\}$. Based on Definitions 1 and 2, we have the following assumptions in NCD.

(A) The support set of $X^l$ and the support set of $X^u$ are disjoint, and underlying classes of $X^l$ are different from those of $X^u$ (i.e., $\mathcal{I} \cap \mathcal{Y} = \emptyset$);

(B) $X^l$ is $K^l$-$\epsilon^l$-separable with $\mathcal{F}^l = \{f^l\}$ and $X^u$ is $K^u$-$\epsilon^u$-separable with $\mathcal{F}^u$, where $\epsilon^l = \tau(X^l, f^l(X^l)) < 1$ and $\epsilon^u = \min_{f \in \mathcal{F}} \tau(X^u, f(X^u)) < 1$;

(C) There exist a consistent $K^l$-$\epsilon^l$-separable transformation set $\Pi^l$ for $X^l$ and a consistent $K^u$-$\epsilon^u$-separable transformation set $\Pi^u$ for $X^u$;

(D) $\Pi^l \cap \Pi^u \neq \emptyset$.

(A) ensures that known and novel classes are disjoint. (B) implies that it is meaningful to separate observations from $X^l$ and $X^u$. (C) means that we can find good high-level features for $X^l$ or $X^u$. Based on these features, it is much easier to separate $X^l$ or $X^u$. (D) says that the high-level features of $X^l$ and $X^u$ are shared, as demonstrated in the introduction. Then, we can define NCD formally.

**Problem 1** (NCD). *Given $X^l$, $X^u$ and $f^l$ defined above and assume (A)-(D) hold, in NCD, we aim to learn a function $\hat{\pi} : \mathcal{X} \to \mathbb{R}^{d_r}$ via minimizing $\mathcal{J}(\hat{\pi}) = \tau(\hat{\pi}(X^l), f^l(X^l)) + \tau(\hat{\pi}(X^u), f^u(X^u))$, where $f^u \in \mathcal{F}$ and $d_r \ll d$. We expect that $\hat{\pi}(X^u)$ is $K^u$-$\epsilon^u$-separable.*

**Theorem 1** (NCD is Theoretically Solvable). *Given $X^l$, $X^u$ and $f^l$ defined above and assume (A)-(D) hold, then $\hat{\pi}(X^u)$ is $K^u$-$\epsilon^u$-separable. If $\epsilon^u = 0$, then NCD is theoretically solvable.*

Theorem 1 means that it is possible to find a good transformation $\hat{\pi}$ such that $\hat{\pi}(X^{\mathrm{u}})$ is separable although we introduce $X^{l1}$. Then we show that NCD might be ill-defined if (**D**) does not hold.

**Theorem 2** (Impossibility Theorem). *Given $X^{\mathrm{l}}$, $X^{\mathrm{u}}$ and $f^{\mathrm{l}}$ defined above and assume (**A**)-(**C**) hold, if $\max_{\pi \in \Pi^{l}} \tau(\pi(X^{\mathrm{u}}), f^{\mathrm{u}}(X^{\mathrm{u}})) < \min_{\pi \in \Pi - \Pi^{l}} \tau(\pi(X^{\mathrm{l}}), f^{\mathrm{l}}(X^{\mathrm{l}}))$, (**D**) does not hold and $\epsilon^{\mathrm{l}} \le \epsilon^{\mathrm{u}}$, then $\tau(\hat{\pi}(X^{\mathrm{u}}), f^{\mathrm{u}}(X^{\mathrm{u}})) > \epsilon^{\mathrm{u}}$, where $\Pi = \{\pi : \mathcal{X} \to \mathbb{R}^{d_r}\}$ and $f^{\mathrm{u}} \in \mathcal{F}$.*

**Remark 2.** Condition $\max_{\pi \in \Pi^{l}} \tau(\pi(X^{\mathrm{u}}), f^{\mathrm{u}}(X^{\mathrm{u}})) < \min_{\pi \in \Pi - \Pi^{l}} \tau(\pi(X^{\mathrm{l}}), f^{\mathrm{l}}(X^{\mathrm{l}}))$ indicates that the worst case of clustering novel classes with transformations that are suitable for known classes is better than the best case of clustering known classes with transformations that are not suitable for known classes. Besides, $\epsilon^{\mathrm{l}} \le \epsilon^{\mathrm{u}}$ indicates that the transformations in $X^{\mathrm{l}}$ is more separable than $X^{\mathrm{u}}$.

In previous setting, data acquiring process is unclear. As we discussed, if NCD is labelling in causality, then novel and known classes might not share high-level semantic features (Figure 1(a)). Namely, (**D**) might not hold, resulting in that $X^{\mathrm{l}}$ might bring negative effects under some conditions (Theorem 2). Thus, based on Theorems 1 and 2, it is clear that (**D**) plays a key role in NCD, which also justifies that NCD will be well-defined if it is sampling in causality. Noting that sampling in causality ($Y \to X$) is the sufficient unnecessary condition of (**D**). For example, if there are some constraints to make the novel-class data (to be annotated) are obtained in the same scenario with known-class data, then data generated by labeling ($X \to Y$) also satisfy (**D**).

Based on Theorem 1, if there are abundant observations to estimate $\mathcal{J}(\hat{\pi})$, then we can find the optimal transformation $\hat{\pi}$ to help obtain a good partition for observations from $X^{\mathrm{u}}$. However, as discussed before, the novel classes might only be seen once or twice in some scenarios. In such scenarios, we find that previous NCD methods do not work well empirically (Figure 2). To address the *NCD given very limited data* (NCDL), we link NCD to meta-learning that also assumes that known and unknown (novel) classes share the high-level semantic features (Chen et al., 2020a), which is exactly the same as (**D**) in NCD. Thus, it is natural to address NCDL by meta-learning.

## 4 META DISCOVERY FOR NCDL

Meta-learning has been widely used to solve few-shot learning problems (Chen et al., 2020a; Wang et al., 2020). The pipeline of meta-learning consists of three steps: 1) randomly sampling data to simulate many inner-tasks; 2) training each inner-task by minimizing its empirical risk; 3) regarding each inner-task as a data point to update meta algorithm. Compared to meta-learning, the inner-task of NCDL is clustering instead of classification in meta-learning. Thus, we can modify the loss function of inner-task to be suitable for clustering and follow the framework of meta-learning, i.e., *meta discovery*. In Appendix C, we give NCDL a definition from the view of meta-learning and further prove NCDL is theoretically solvable. In this section, we let $S^{\mathrm{l}} = \{(\boldsymbol{x}_i^{\mathrm{l}}, y_i^{\mathrm{l}}) : i = 1, \ldots, n^{\mathrm{l}}\}$ be known-class data drawn from r.v. $(X^{\mathrm{l}}, f^l(X^{\mathrm{l}}))$ and $y_i^{\mathrm{l}} \in \{1, \ldots, K^{\mathrm{l}}\}$, and let $S^{\mathrm{u}} = \{\boldsymbol{x}_i^{\mathrm{u}} : i = 1, \ldots, n^{\mathrm{u}}\}$ be unlabeled novel-class data drawn from r.v. $X^{\mathrm{u}}$, where $0 < n^{\mathrm{u}} \ll n^{\mathrm{l}}$.

Due to the difference of inner-task, if we randomly sample data to compose an inner-task like existing meta-learning methods, these data may negatively influence each other in the training procedure. This is because these randomly sampled data in an inner-task have different clustering rules (Figure 3). Thus, in meta discovery, the key is to propose a new task sampler that takes care of clustering rules.

**CATA: Clustering-rule-aware Task Sampler.** From the perspective of multi-view learning (Blum & Mitchell, 1998), data usually contain different feature representations. Namely, data have multiple views. However, there are always one view or a few views that are dominated for each instance, and these dominated views are similar with high-level semantic meaning (Li et al., 2019b; Liu et al., 2020d; 2021a). Therefore, we propose to use dominated views to replace with clustering rules, and design a novel task-sampling method called *clustering-rule-aware task sampler* (CATA, Algorithm 1). CATA is based on a multi-view network containing a feature extractor $G : \mathcal{X} \to \mathbb{R}^{d_s}$ and $M$ classifiers $\{F_i : \mathbb{R}^{d_s} \to \mathcal{Y}\}_{i=1}^M$ (Figure 4). $M$ is empirically chosen according to the data complexity. Specifically, CATA learns a low-dimension projection and set of orthogonal classifiers. It then assigns each data point to a group defined by which of the orthogonal classifiers was most strongly activated by the observation, and inner tasks are sampled from each group.

---

[1]Note that, the objective of clustering problem is different from that of NCD. In clustering problem, we aim to find $\hat{\pi}$ via minimizing $\tau(\hat{\pi}(X^{\mathrm{u}}), f^{\mathrm{u}}(X^{\mathrm{u}}))$ rather than $\tau(\hat{\pi}(X^{\mathrm{l}}), f^{\mathrm{l}}(X^{\mathrm{l}})) + \tau(\hat{\pi}(X^{\mathrm{u}}), f^{\mathrm{u}}(X^{\mathrm{u}}))$.

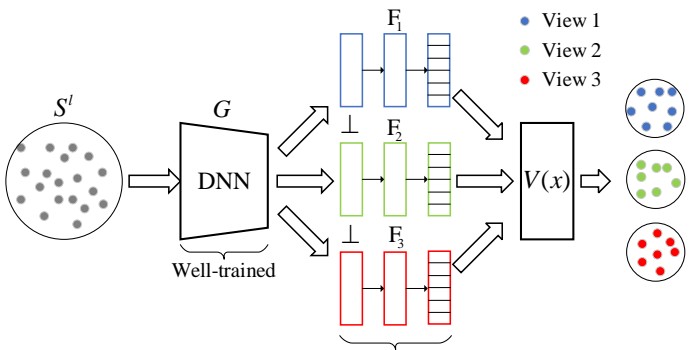

CATA is a novel sampling method of meta-learning for NCDL. Here we show the inference process of assigning labeled data of known classes to three different views with the well-trained $G$ and $\{F_i\}_{i=1}^3$. $V(x)$ is the voting function defined as Eq. (4). The weights of the first layers of $F_1$, $F_2$, and $F_3$ are constrained to orthogonal mutually.

Figure 4: The structure of the *clustering-rule-aware task sampler* (CATA).

The feature extractor $G$ provides the shared data representations for $M$ different classifiers $\{F_i\}_{i=1}^M$. Each classifier classifies data based on its own view. The feature extractor $G$ learns from all gradients from $\{F_i\}_{i=1}^M$. To ensure that different classifiers have different views, we constrain the weight vector of the first fully connected layer of each classifier to be orthogonal. Take $F_i$ and $F_j$ as an example, we add the term $|W_i^T W_j|$ to the sampler's loss function, where $W_i$ and $W_j$ denote the weight vectors of the first fully connected layer of $F_i$ and $F_j$ respectively. $|W_i^T W_j|$ tending to 0 means that $F_i$ and $F_j$ are nearly independent (Saito et al., 2017). Thus, the loss function of CATA is defined as follows,

$$\mathcal{L}_S(\theta_G, \{\theta_{F_i}\}_{i=1}^M) = \frac{1}{MN} \sum_{j=1}^M \sum_{i=1}^N \ell_{ce}(F_j \circ G(x_i), y_i) + \frac{2\lambda}{M(M-1)} \sum_{i \neq j} |W_i^T W_j|, \quad (3)$$

where $\ell_{ce}$ is the standard cross-entropy loss function and $\lambda$ is a trade-off parameter.

After we obtain the well-trained feature extractor $G$ and classifiers $\{F_i\}_{i=1}^M$, we input a training data point $x$ to our sampler and then we will get the probabilities that $x$ belongs to class $y$ in each classifier, i.e. $\{P_i(y|x)\}_{i=1}^M$, where $y$ is the label of $x$. Therefore, the view which $x$ belongs to is defined as

$$V(x) = \arg\max_i P_i(y|x). \quad (4)$$

Now we have assigned a data point to $M$ subsets according to their views, i.e. $\{V_i = \{x \in \mathcal{X} : V(x) = i\}\}_{i=1}^M$. Then, we can directly randomly sample a certain number of data (e.g., $N$-way, $K$-observation) from one subset to compose an inner task. According to the number of data in each subset, we sample inner tasks from each subset with different frequencies. We also compare CATA with commonly used samplers in meta-learning in Appendix A. Note that, CATA is a heuristic method, and we will give clustering rule a formal definition and explore the reason why CATA succeed theoretically in the future.

**Realization of Meta Discovery with MAML (MM).** Here, we solve the NCDL problem based on MAML. A feature extractor $\pi_{mm} : \mathcal{X} \to \mathbb{R}^{d_r}$ is given to obtain an embedding of data, following a classifier $g$ with the output dimension equalling to the number of novel classes ($K^u$). As novel classes share the same high-level semantic features of known classes, the feature extractor $\pi_{mm}$ should be applicable to known and novel classes. The key idea is that similar data should belong to the same class. For data pair $(x_i, x_j)$, let $s_{ij} = 1$ if they come from the same class; otherwise, $s_{ij} = 0$.

Following Han et al. (2020b), we adopt a more robust pairwise similarity called *ranking statistics*. For $z_i = \pi_{mm}(x_i)$ and $z_j = \pi_{mm}(x_j)$, we rank the values of $z_i$ and $z_j$ by the magnitude. Then we check if the indices of the values of top-$k$ ranked dimensions are the same. Namely, $s_{ij} = 1$ if they are the same, and $s_{ij} = 0$ otherwise. We use the pairwise similarities $\{s_{ij}\}_{1 \leq i,j \leq n^l}$ as pseudo labels to train feature extractor $\pi_{mm}$ and classifier $g$. As mentioned above, $g$ is a classifier with softmax layer, so the inner product $g(z_i)^T g(z_j)$ is the cosine similarity between $x_i$ and $x_j$, which serves as the score for whether $x_i$ and $x_j$ belong to the same class. After sampling training tasks $\{\mathcal{T}_i\}_{i=1}^n$, we train a model by the inner algorithm that optimizes the *binary cross-entropy* (BCE) loss function:

$$\mathcal{L}_{\mathcal{T}_i}(\theta_{g \circ \pi_{mm}}) = -\frac{1}{n^{l2}} \sum_{i=1}^{n^l} \sum_{j=1}^{n^l} [s_{ij} \log(g(z_i)^T g(z_j)) + (1 - s_{ij}) \log(1 - g(z_i)^T g(z_j))]. \quad (5)$$

---

**Algorithm 1** Clustering-rule-aware task sampler (CATA)

---

**Input:** known-class data $S^l$, feature extractor $G$, classifiers $\{F_i\}_{i=1}^{M^l}$, learning rates $\omega_1, \omega_2$; view index: $j$.

**1: Initialize** $\theta_G$ and $\{\theta_{F_i}\}_{i=1}^{M^l}$;

**for** $t = 1, \ldots, T$ **do**

    **2: Compute** $\nabla_{\theta_G}\mathcal{L}_S$ and $\{\nabla_{\theta_{F_i}}\mathcal{L}_S\}_{i=1}^{M^l}$ using $S^l$ and $\mathcal{L}_S$ in Eq. (3);

    **3: Update** $\theta_G = \theta_G - \omega_1\nabla_{\theta_G}\mathcal{L}_S(\theta_G, \{\theta_{F_i}\}_{i=1}^{M^l}), \theta_{F_i} = \theta_{F_i} - \omega_2\nabla_{\theta_{F_i}}\mathcal{L}_S(\theta_G, \{\theta_{F_i}\}_{i=1}^{K^l}), i = 1, \ldots, M^l$;

**end**

**4: Compute** $\{F_i(G(\boldsymbol{x}^l))\}_{i=1}^{M^l}$ to obtain $\{P_i(y^l|\boldsymbol{x}^l)\}_{i=1}^{M^l}$ for each $(\boldsymbol{x}^l, y^l) \in S^l$;

**5: Compose** $V_i = \{\boldsymbol{x}^l : V(\boldsymbol{x}^l) = i\}$ using function $V$ in Eq. (4), $i = 1, \ldots, M^l$;

**6: Sample** an inner-task $\mathcal{T}_i = (S_i^{l,tr}, S_i^{l,ts}) \sim V_j$

**Output:** an inner-task $\mathcal{T}_i$

---

---

**Algorithm 2** MM for NCDL.

---

**Input:** known-class data: $S^l$; learning rate: $\alpha, \eta$; feature extractor: $\pi_{mm}$; classifier: $g$

**1: Initialize** $\theta_{g\circ\pi_{mm}}$;

**while** *not done* **do**

    **2: Sample tasks** $\{\mathcal{T}_i = (S_i^{l,tr}, S_i^{l,ts}) \sim S^l\}_{i=1}^n$ by **CATA** (Alg. 1);

    **for** *all* $\mathcal{T}_i$ **do**

        **3: Evaluate** $\nabla_{\theta_{g\circ\pi_{mm}}}\mathcal{L}_{\mathcal{T}_i}(\theta_{g\circ\pi_{mm}})$ using $S_i^{l,tr}$ and $\mathcal{L}_{\mathcal{T}_i}$ in Eq. (5);

        **4: Compute** adapted parameters: $\theta'_{g\circ\pi_{mm},i} = \theta_{g\circ\pi_{mm}} - \alpha\nabla_{\theta_{g\circ\pi_{mm}}}\mathcal{L}_{\mathcal{T}_i}(\theta_{g\circ\pi_{mm}})$;

    **end**

    **5: Update** $\theta_{g\circ\pi_{mm}} = \theta_{g\circ\pi_{mm}} - \eta\nabla_{\theta_{g\circ\pi_{mm}}}\mathcal{L}_A(\theta'_{g\circ\pi_{mm}})$ using each $S_i^{l,ts}$ and $\mathcal{L}_A$ in Eq. (6);

**end**

**Output:** clustering algorithm $\bar{A}$.

---

Entire procedures of NCDL by MAML are shown in Algorithm 2. Following MAML, the parameters of clustering algorithm $\boldsymbol{A}$ are trained by optimizing the following loss function:

$$\mathcal{L}_{\boldsymbol{A}}(\theta_{g\circ\pi_{mm}}) = \sum_{i=1}^n \mathcal{L}_{\mathcal{T}_i}(\theta_{g\circ\pi_{mm}} - \alpha\nabla_{\theta_{g\circ\pi_{mm}}}\mathcal{L}_{\mathcal{T}_i}(\theta_{g\circ\pi_{mm}})), \tag{6}$$

where $\alpha > 0$ is the learning rate of the inner-algorithm. Then we conduct the meta-optimization to update the parameters of clustering algorithm $\boldsymbol{A}$ as follows:

$$\theta_{g\circ\pi_{mm}} \leftarrow \theta_{g\circ\pi_{mm}} - \eta\nabla_{\theta_{g\circ\pi_{mm}}}\mathcal{L}_{\boldsymbol{A}}(\theta_{g\circ\pi_{mm}}), \tag{7}$$

where $\eta > 0$ denotes the meta learning rate. After finishing meta-optimization, we finetune the clustering algorithm $\boldsymbol{A}$ with the novel-class data to yield a new clustering algorithm that is adapted to novel classes. More specifically, we perform line 4 and line 5 in Algorithm 2 with $S^u$.

**Realization of Meta Discovery with ProtoNet (MP).** Following Snell et al. (2017), we denote $\pi_{mp}$ as a feature extractor, which maps data to their representations. In training task $\mathcal{T}_i$, the mean vector of representations of data from class-$s$ (i.e., $S_{i,s}^{l,tr}$) is defined as prototype $\boldsymbol{c}_k$:

$$\boldsymbol{c}_{i,s}(S_{i,s}^{l,tr}) = \frac{1}{|S_{i,s}^{l,tr}|}\sum_{(x_i^l, y_i^l)\in S_{i,s}^{l,tr}}\pi_{mp}(x_i^l). \tag{8}$$

Here, we define the Euclidean distance $dist : \mathbb{R}^{d_r} \times \mathbb{R}^{d_r} \to [0, +\infty)$ to measure the distance between data from the test set and the prototype. Then, we represent $p(y = s|\boldsymbol{x})$ using the following equation.

$$p(y = s|\boldsymbol{x}) = \frac{\exp(-dist(\pi_{mp}(\boldsymbol{x}), \boldsymbol{c}_s))}{\sum_{s'}\exp(-dist(\pi_{mp}(\boldsymbol{x}), \boldsymbol{c}_{s'}))}. \tag{9}$$

We train the feature extractor by optimizing the negative log-probability, i.e., $-\log p(y = s|\boldsymbol{x})$. So the loss function of ProtoNet is defined as follows:

$$\mathcal{L}_{\mathcal{T}_i} = -\frac{1}{k}\sum_{s\in[K^u]}\sum_{\boldsymbol{x}\in S_{i,s}^{l,ts}}\log p(y = s|\boldsymbol{x}), \tag{10}$$

---

**Algorithm 3** MP for NCDL.

---

**Input:** known-class data: $S^{\mathrm{l}}$; learning rate: $\gamma$; feature extractor: $\pi_{\mathrm{mp}}$
**1: Initialize** $\theta_{\pi_{\mathrm{mp}}}$;
**while** *not done* **do**
    **for** *all episodes* **do**
        **2: Sample** $K^{\mathrm{u}}$ elements from $\{1, \ldots, K^{\mathrm{l}}\}$ as set $C_I$;
        **3: Sample tasks** $\mathcal{T}_i = (S_i^{\mathrm{l,tr}}, S_i^{\mathrm{l,ts}}) \sim S^{\mathrm{l}}{}_{|y^{\mathrm{l}} \in C_I}$ by **CATA** (Alg. 1);
        **for** *s in* $C_I$ **do**
            **4: Sample** $m$ training data of class-$s$, i.e., $S_{i,s}^{\mathrm{l,tr}} \sim S_i^{\mathrm{l,tr}}$ & $|S_{i,s}^{\mathrm{l,tr}}| = m$;
            **5: Compute** $c_{i,s}(S_{i,s}^{\mathrm{l,tr}})$ using Eq. (8);
            **6: Sample** $k$ test data of class-$s$, i.e., $S_{i,s}^{\mathrm{l,ts}} \sim S_i^{\mathrm{l,ts}}$ & $|S_{i,s}^{\mathrm{l,ts}}| = k$;
        **end**
        **7: Update** $\theta_{\pi_{\mathrm{mp}}} = \theta_{\pi_{\mathrm{mp}}} - \gamma \nabla_{\theta_{\pi_{\mathrm{mp}}}} \mathcal{L}_{\mathcal{T}_i}(\theta_{\pi_{\mathrm{mp}}})$ using $\{S_{i,s}^{\mathrm{l,ts}}\}_{s \in C_I}$ and $\mathcal{L}_{\mathcal{T}_i}$ in Eq. (10);
    **end**
**end**
**Output:** feature extractor $\pi_{\mathrm{mp}}$.

---

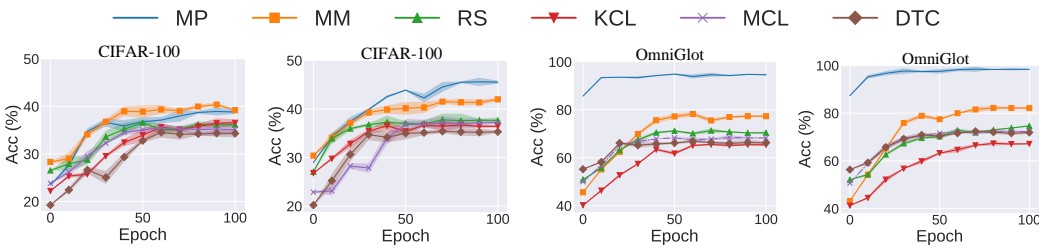

(a) 20-way 1-observation  (b) 20-way 5-observation  (c) 20-way 1-observation  (d) 20-way 5-observation

Figure 5: We conducted experiments on CIFAR-100 and OmniGlot and reported the *average clustering accuracy* (ACC (%), Section 5) when using existing and our methods to address the NCDL problem. The experimental results showed that MM and MP tend to outperform existing methods.

where $[K^{\mathrm{u}}]$ denotes the $K^{\mathrm{u}}$ classes selected from $\{1, \ldots, K^{\mathrm{l}}\}$. $S_{i,s}^{\mathrm{l,ts}}$ is the test set of labeled data of class-$s$ from task $\mathcal{T}_i$. The full procedures of training $\pi_{\mathrm{mp}}$ are shown in Algorithm 3. After training the feature extractor $\pi_{\mathrm{mp}}$ well, we use the training set of $S^{\mathrm{u}}$ to obtain the prototypes. For $x$ in the test set of $S^{\mathrm{u}}$, we compute the distance between $x$ and each prototype, and then the class corresponding to the nearest prototype is the class of $x$.

## 5 EXPERIMENTS

**Baselines.** To verify the performance of our meta-based NCDL methods (i.e., MM and MP), we compare them with 5 competitive baselines, including K-means (MacQueen et al., 1967), KCL (Hsu et al., 2018) , MCL (Hsu et al., 2019), DTC (Han et al., 2019), and RS (Han et al., 2020b). We modify these baselines by only reducing the amount of novel-class data, with other configurations invariable. We clarify the implementation details of CATA, MM, and MP in Appendix E.

**Datasets.** To evaluate the performance of our methods and baselines, we conduct experiments on four popular image classification benchmarks, including CIFAR-10 (Krizhevsky & Hinton, 2009), CIFAR-100 (Krizhevsky & Hinton, 2009), SVHN (Netzer et al., 2011), and OmniGlot (Lake et al., 2015). Detailed introductions and partitions of known classes and novel classes of these four datasets is in Appendix D. Following the protocol of few-shot learning (Park et al., 2018; Liu et al., 2019b; Wang et al., 2020; Ziko et al., 2020), for SVHN and CIFAR-10, we perform the few-observation tasks of 5-way 1-observation and 5-way 5-observation, and we perform the few-observation tasks of 20-way 1-observation and 20-way 5-observation for CIFAR-100 and OmniGlot.

**Evaluation metric.** For a clustering problem, we use the *average clustering accuracy* (ACC) to evaluate the performance of clustering, which is defined as follows,

$$\max_{\phi \in L} \frac{1}{N} \sum_{i=1}^{N} \mathbb{1}\{\bar{y}_i = \phi(y_i)\}, \tag{11}$$

Table 1: Ablation Study on four datasets. In this table, we report the ACC (%)±standard deviation of ACC (%) on four datasts, where w/o represents "without". It is clear that CATA improves the ACC.

| Methods | | MM | MM w/o CATA | MP | MP w/o CATA |
|---|---|---|---|---|---|
| SVHN (5-way) | 5-observation | 47.3±0.3 | 40.1±0.3 | 61.0±0.5 | 60.5±0.2 |
| | 1-observation | 42.7±0.3 | 39.7±0.4 | 52.8±0.4 | 50.2±0.3 |
| CIFAR-10 (5-way) | 5-observation | 45.3±0.1 | 42.7±0.3 | 58.5±0.2 | 57.9±0.1 |
| | 1-observation | 41.3±0.4 | 40.3±0.3 | 51.7±0.2 | 47.6±0.2 |
| CIFAR-100 (20-way) | 5-observation | 42.0±0.4 | 39.9±0.3 | 45.5±0.3 | 44.1±0.1 |
| | 1-observation | 39.2±0.2 | 37.1±0.3 | 38.8±0.4 | 37.0±0.2 |
| OmniGlot (20-way) | 5-observation | 82.1±0.4 | 80.1±0.4 | 98.4±0.2 | 96.7±0.3 |
| | 1-observation | 77.3±0.4 | 78.5±0.4 | 94.6±0.3 | 91.2±0.2 |

where $\bar{y}_i$ and $y_i$ denote the ground-truth label and assigned cluster indices respectively. $L$ is the set of mappings from cluster indices to ground-truth labels.

**Results on CIFAR-10.** As shown in Figures 2(a) and 2(b), MM and MP outperform all baselines significantly, and the ACC of MP is much higher than that of MM. The main reason is that MP makes full use of the labels of known-class data in the training process, while MM does not. MM only uses the labels of known-class data in the sampling process. Besides, as shown in Table 2 in Appendix F, K-means performs better on CIFAR-10 than other datasets. The reason is that clustering rules contained in data of CIFAR-10 are simple and suitable for K-means.

**Results on SVHN.** Figures 2(d) and 2(c) show that our methods still outperform all baselines. In the task of 5-way 1-observation, RS performs as well as MM (Figure 2(d)). The reason is that RS trains the embedding network with self-supervised learning method, RotationNet (Gidaris et al., 2018), under 1-observation case, which partly overcomes this problem by data augment. The SVHN is simpler than other datasets, thus such a data augmentation works better.

**Results on CIFAR-100.** It is clear that we outperform all baselines. Differ from tasks on other datasets, MM performs equally even a little better than MP shown in Figure 5(a). The reason is that the amount of known classes is relatively large and the data distribution of CIFAR-100 is complex, so we cannot accurately compute prototypes with very limited data.

**Results on OmniGlot.** As shown in Figures 5(c) and 5(d), our methods still have the highest ACC. We find that the ACC of K-means are merely 2.2% for both 1-observation and 5-observation, indicating that K-means hardly works on OmniGlot. Although this result looks very bad, this is reasonable. This is because the number of novel classes is too large (i.e., 659) and K-means is an unsupervised method that requires many training data. As a simple benchmark in few-shot learning, existing meta-learning methods (Ramalho & Garnelo, 2019; Li et al., 2019a) have completed solved it, which achieved the accuracy of 99.9% on 20-way 5-shot task. Thus our method MP also achieves a high accuracy of 98.4% without novel-class labels. Note that, we also show results of all methods on NCD problem in Table 3 (Appendix G).

**Ablation study.** To verify the effectiveness of CATA, we conduct ablation study by removing CATA from MM and MP. According to Table 1, CATA significantly improves the performance of MM and MP. However, there exists an abnormal phenomenon in OmniGlot, i.e., MM w/o CATA outperforms MM in the task of 20-way 1-observation. Although we need $16 (= |S^{l,tr}| + |S^{l,ts}| = 1 + 15)$ data for each class in an inner-task, the total amount of data for each class is only 20. Therefore, there are not enough data for CATA to sample, which makes CATA cannot improve the ACC of MM.

## 6 CONCLUSIONS

In this paper, we study an important problem called *novel class discovery* (NCD) and demystify the key assumptions behind this problem. We find that NCD is sampling instead of labeling in causality, and, furthermore, data in the NCD problem should share high-level semantic features. This finding motivates us to link NCD to meta-learning, since meta-learning also assumes that the high-level semantic features are shared between seen and unseen classes. To this end, we propose to discover novel classes in a meta-learning way, i.e., the meta discovery. Results show that meta-learning based methods outperform all existing baselines when addressing a more challenging problem *NCD given very limited data* (NCDL) where only few novel-class data can be observed, which lights up a novel road for NCD/NCDL.

## 7 ACKNOWLEDGEMENTS

This work was partially supported by the National Natural Science Foundation of China (No. 91948303-1, No. 61803375, No. 12002380, No. 62106278, No. 62101575, No. 61906210) and the National Grand R&D Plan (Grant No. 2020AAA0103501). BH was supported by NSFC Young Scientists Fund No. 62006202 and RGC Early Career Scheme No. 22200720. TLL was supported by Australian Research Council Projects DE-190101473 and DP-220102121. MS was supported by JST CREST Grant Number JPMJCR18A2.

## 8 ETHICS STATEMENT

This paper does not raise any ethics concerns. This study does not involve any human subjects, practices to data set releases, potentially harmful insights, methodologies and applications, potential conflicts of interest and sponsorship, discrimination/bias/fairness concerns, privacy and security issues, legal compliance, and research integrity issues.

## 9 REPRODUCIBILITY STATEMENT

To ensure the reproducibility of experimental results, we have provided codes of MM and MP at github.com/Haoang97/MEDI.

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

# A    DETAILED RELATED WORK

**Novel class discovery.** NCD is proposed in recent years, aiming to cluster unlabeled novel-class data according to their underlying categories (Han et al., 2021; Jia et al., 2021). Compared with unsupervised learning (Barlow, 1989), NCD also requires labeled known-class data to help cluster novel-class data. The pioneering methods include *KLD-based contrastive loss* (KCL) (Hsu et al., 2018), *meta classification likelihood* (MCL) (Hsu et al., 2019), *deep transfer clustering* (DTC) (Han et al., 2019), *rank statistics* (RS) (Han et al., 2020b), OpenMix (Zhong et al., 2021c), *neighborhood contrastive learning* (NCL) (Zhong et al., 2021a), and *unified objective* (UNO) (Fini et al., 2021). In this paper, we update and present the results of our methods regarding NCD.

In KCL (Hsu et al., 2018), a method based on pairwise similarity is introduced. They first pre-trained a similarity prediction network on labeled data of known classes and then use this network to predict the similarity of each unlabeled data pair, which acts as the supervision information to train the main model. Then, MCL (Hsu et al., 2019) changed the loss function of KCL (the KL-divergence based contrastive loss) to the meta classification likelihood loss.

In DTC (Han et al., 2019), they first learned a data embedding with metric learning on labeled data, and then they employed the DEC (Xie et al., 2016) to learn the cluster assignments on unlabeled data.

In RS (Han et al., 2020b), they used the rank statistics to predict the pairwise similarity of data. To keep the performance on data of known classes, they pre-trained the data embedding network with self-supervised learning method (Gidaris et al., 2018) on both labeled data and unlabeled data.

In OpenMix (Zhong et al., 2021c), they proposed to mix known-class and novel-class data to learn a joint label distribution, benefiting to find their finer relations.

In NCL (Zhong et al., 2021a), they used neighborhood contrastive learning to learn discriminate features with both the labeled and unlabeled data with the local neighborhood to take the knowledge from more positive samples. In addition, they used the hard negative generation to produce hard negative to improve NCL.

In UNO (Fini et al., 2021), they used pseudo-labels in combination with ground-truth labels in a *UNified Objective function* (UNO) that enabled better cooperation and less interference without self-supervised learning.

**Meta-Learning.** Meta-learning is also known as learning-to-learn, which train a meta-model over a large variety of learning tasks (Ravi & Larochelle, 2017; Liu et al., 2019b; 2021b). In meta-learning, we often assume that data share the same high-level features, which ensures that meta-learning can be theoretically addressed (Maurer, 2005). According to (Hospedales et al., 2020), there are three common approaches to meta-learning: optimization-based (Finn et al., 2017), model-based (Santoro et al., 2016), and metric-based (Snell et al., 2017).

Optimization-based methods include those where the inner-level task is literally solved as an optimization problem, and focus on extracting meta knowledge required to improve optimization performance. In model-based methods, the inner learning step is wrapped up in the feed-forward pass of a single model. Metric-based methods perform non-parametric learning at the inner-task level by simply comparing validation points with training points and predicting the label of matching training points. Since meta-learning and the NCDL have the same assumption that data share the same high-level semantic features (introduced in Section 1), we link NCDL to meta-learning problem, providing a way to formulate and analyze the NCDL.

**Positive-unlabeled learning.** *Positive-unlabeled* (PU) learning (Li & Liu, 2005) is an important branch of semi-supervised learning, aiming to learn a binary classifier with positive data and unlabeled data. Thus, PU learning is a special case of NCD, where there exists only one known class and one novel class. The basic solution is to view unlabeled data as negative data to train a standard classifier (Elkan & Noto, 2008). Niu et al. (2016) gives the conditions when PU learning outperforms supervised learning through upper bounds on estimation errors. Kiryo et al. (2017) proposes a non-negative risk estimator to prevent flexible models overfitting on negative data in PU learning.

**Transfer learning.** Transfer learning aims to leverage knowledge contained in source domains to improve the performance of tasks in a target domain, where both domains are similar but different (Gong et al., 2016; Long et al., 2018; Zhou et al., 2019; Liu et al., 2019a; Wang et al., 2019; Liu et al.,

2020c; Liang et al., 2020; Chi et al., 2021; Fang et al., 2021a;b). Representative transfer learning works are domain adaptation (Gong et al., 2016; Long et al., 2018; Zhou et al., 2019; Liu et al., 2019a; 2020c; Dong et al., 2020; 2021b;a) and hypothesis transfer (Kuzborskij & Orabona, 2013; Liang et al., 2020; Chi et al., 2021), which mainly focus on classification or prediction tasks in the target domain. NCD problem can be also regarded a transfer learning problem that aims to complete the clustering task in a target domain via leveraging knowledge in source domains.

**Compared to samplers in meta-learning.** Tasks in meta-learning are heterogeneous in some scenarios, which can not be handled via globally sharing knowledge among data. Therefore, it is crucial to address the task-sampling problem in meta-learning. Yao et al. (2019) assigned many tasks that are randomly sampled from different clusters using their similarities, and only used the most related task cluster for training. This method solves the task-sampling problem *in the view of tasks*. Liu et al. (2020a) proposed a greedy class-pair based sampling method, which selects difficult tasks according to the class-pair potentials. This method solves the sampling problem *in the view of classes*. In our paper, we propose CATA based on clustering rules regarding data, which is *in the view of data*.

# B   MAIN THEORETICAL RESULTS

**Theorem 1** (NCD is Theoretically Solvable). *Given $X^l$, $X^u$ and $f^l$ defined above and assume (A)-(D) hold, then $\hat{\pi}(X^u)$ is $K^u$-$\epsilon^u$-separable. If $\epsilon^u = 0$, then NCD is theoretically solvable.*

*Proof.* The key to this proof is that the optimized $\hat{\pi}^*$ is in $\Pi^l \cap \Pi^u$.

**Case1.** When $f^u \in \mathcal{F}^u$, according to Problem 1, let

$$\hat{\pi}^* = \arg\min_{\pi \in \Pi} \tau(\hat{\pi}(X^l), f^l(X^l)) + \tau(\hat{\pi}(X^u), f^u(X^u)), \qquad (12)$$

where $\Pi = \{\pi : \mathcal{X} \to \mathbb{R}^{d_r}\}$ and $f^u \in \mathcal{F}^u$. Let $\tau^* = \min_{\pi \in \Pi, f^u \in \mathcal{F}^u} \mathcal{J}(\pi)$. If $\hat{\pi}^* \notin \Pi^l \cap \Pi^u$, then, according to definitions of $\Pi^l$ and $\Pi^u$, $\tau^* > \epsilon^l + \epsilon^u$. This means that there exists $\pi' \in \Pi^l \cap \Pi^u$ such that $\mathcal{J}(\pi') = \tau(\pi'(X^l), f^l(X^l)) + \tau(\pi'(X^u), f^u(X^u)) = \epsilon^l + \epsilon^u < \tau^*$. Namely, $\tau^*$ is not the minimum value in the set $\{\mathcal{J}(\pi) : \pi \in \Pi\}$, which leads to a contradiction to the definition of $\tau^*$.

**Case2.** When $f^u \notin \mathcal{F}^u$, let $\tau^{**} = \min_{\pi \in \Pi, f^u \notin \mathcal{F}^u} \mathcal{J}(\pi)$. According to definition of $\epsilon^u$, it is clear that $\tau^{**} > \epsilon^l + \epsilon^u = \mathcal{J}(\pi')$. Namely, $\tau^{**}$ is not the minimum value in the set $\{\mathcal{J}(\pi) : \pi \in \Pi\}$.

**NCD is solvable.** If $\epsilon^u = 0$, according to definition of $\Pi^u$, $\tau(\hat{\pi}^*(X^u), f^u(X^u)) = 0$, which means that we can perfectly separate $\hat{\pi}^*(X^u)$. Namely, NCD is theoretically solvable. □

**Theorem 2** (Impossibility Theorem). *Given $X^l$, $X^u$ and $f^l$ defined above and assume (A)-(C) hold, if $\max_{\pi \in \Pi^l} \tau(\pi(X^u), f^u(X^u)) < \min_{\pi \in \Pi - \Pi^l} \tau(\pi(X^l), f^l(X^l))$, (D) does not hold and $\epsilon^l \leq \epsilon^u$, then $\tau(\hat{\pi}(X^u), f^u(X^u)) > \epsilon^u$, where $\Pi = \{\pi : \mathcal{X} \to \mathbb{R}^{d_r}\}$ and $f^u \in \mathcal{F}$.*

*Proof.* If $f^u \notin \mathcal{F}^u$, then we naturally have $\tau(\hat{\pi}(X^u), f^u(X^u)) > \epsilon^u$ according to the definition of $\epsilon^u$. Then, the key to this proof is that the optimized $\hat{\pi}^*$ is in $\Pi^l$. According to Problem 1,

$$\hat{\pi}^* = \arg\min_{\pi \in \Pi} \tau(\hat{\pi}(X^l), f^l(X^l)) + \tau(\hat{\pi}(X^u), f^u(X^u)). \qquad (13)$$

Let $\tau^* = \min_{\pi \in \Pi} \mathcal{J}(\pi)$. If $\hat{\pi}^* \in \Pi - \Pi^l$, then, according to definition of $\Pi^u$, $\epsilon^u \geq \epsilon^l$, and $\max_{\pi \in \Pi^l} \tau(\pi(X^u), f^u(X^u)) < \min_{\pi \in \Pi - \Pi^l} \tau(\pi(X^l), f^l(X^l))$, we have the following inequality.

$$\tau^* = \mathcal{J}(\hat{\pi}^*) \geq \tau(\hat{\pi}^*(X^l), f^l(X^l)) + \epsilon^u > \max_{\pi \in \Pi^l} \tau(\pi(X^u), f^u(X^u)) + \epsilon^l. \qquad (14)$$

Let $\pi' = \arg\max_{\pi \in \Pi^l} \tau(\pi(X^u), f^u(X^u))$. Since $\pi' \in \Pi^l$, we know that $\tau(\pi'(X^l), f^l(X^l)) = \epsilon^l$ according to definition of $\Pi^l$. Thus we have

$$\max_{\pi \in \Pi^l} \tau(\pi(X^u), f^u(X^u)) + \epsilon^l = \tau(\pi'(X^u), f^u(X^u)) + \tau(\pi'(X^l), f^l(X^l)) = \mathcal{J}(\pi'). \qquad (15)$$

Hence, we find a $\pi' \in \Pi$ such that $\mathcal{J}(\pi') < \mathcal{J}(\hat{\pi}^*) = \tau^*$, which leads to a contradiction to the definition of $\tau^*$. Thus, $\hat{\pi}^* \in \Pi^l$. Based on the definition of $\Pi^u$, $\tau(\hat{\pi}(X^u), f^u(X^u)) > \epsilon^u$. □

## C  NCDL IN THE VIEW OF META-LEARNING

Since we address NCDL based on the meta-learning framework, it is interesting to analyze if NCDL can be addressed based on the sampled tasks $\boldsymbol{T} = \{\mathcal{T}_i\}_{i=1}^n$. We show that, under some assumptions, NCDL can be well addressed.

**Problem Setup of NCDL in the View of Meta-learning.** In NCDL, we have a task space $\mathcal{T}^* = \{(X, f) : X \text{ is any r.v.s defined on } \mathcal{X}, f : \mathcal{X} \to \mathcal{C}\}$, a task distribution $\mathcal{P}(\mathcal{T}^*)$ defined on $\mathcal{T}^*$, a r.v. $X^{\mathrm{u}}$, sampled tasks $\mathcal{T}_i = \{X_i^{\mathrm{l}}, f_i^{\mathrm{l}}\} \sim \mathcal{P}(\mathcal{T}^*)$ $(i = 1, \ldots, n)$, an index set $\mathcal{I} = \{i_1^{\mathrm{u}}, \ldots, i_{K^{\mathrm{u}}}^{\mathrm{u}}\}$, a function set $\mathcal{F} = \{f : \mathcal{X} \to \mathcal{C}\}$, a transformation set $\Pi = \{\pi : \mathcal{X} \to \mathbb{R}^{d_r}\}$, and the loss function $\ell : \mathbb{R}^{d_r} \times \mathcal{F} \to \mathbb{R}^+$ that is the loss function such that, $\forall f \in \mathcal{F}$ and $\forall \pi \in \Pi$, $\mathbb{E}_{\mathbb{P}_X}[\ell(\pi(X), f(X))] = \tau(\pi(X), f(X))$, where $f^{\mathrm{l}} : \mathcal{X} \to \mathcal{Y}$ is the ground-truth labeling function for $X_i^{\mathrm{l}}$, $\mathcal{Y} = \{i_1^{\mathrm{l}}, \ldots, i_{K^{\mathrm{l}}}^{\mathrm{l}}\}$, $\mathcal{C} = \mathcal{Y} \cup \mathcal{I}$ and $\mathbb{P}_X$ is the distribution corresponding to a r.v. $X$. Based on Definitions 1 and 2, we have the following assumptions in NCDL.

(A1) The union of support set of $X_i^{\mathrm{l}}$ $(i = 1, \ldots, n)$ and the support set of $X^{\mathrm{u}}$ are disjoint, and union of underlying classes of $X_i^{\mathrm{l}}$ $(i = 1, \ldots, n)$ are different from those of $X^{\mathrm{u}}$;

(B1) $X_i^{\mathrm{l}}$ is $K^{\mathrm{l}}$-$\epsilon_i^{\mathrm{l}}$-separable with $\mathcal{F}^{\mathrm{l}} = \{f_i^{\mathrm{l}}\}$ and $X^{\mathrm{u}}$ is $K^{\mathrm{u}}$-$\epsilon^{\mathrm{u}}$-separable with $\mathcal{F}^{\mathrm{u}}$, where $\epsilon_i^{\mathrm{l}} = \tau(X_i^{\mathrm{l}}, f_i^{\mathrm{l}}(X_i^{\mathrm{l}})) < 1$, $\epsilon^{\mathrm{u}} = \min_{f \in \{f : \mathcal{X} \to \mathcal{I}\}} \tau(X^{\mathrm{u}}, f(X^{\mathrm{u}})) < 1$;

(C1) There exist a consistent $K^{\mathrm{l}}$-$\epsilon_i^{\mathrm{l}}$-separable transformation set $\Pi_i^{\mathrm{l}}$ for $X_i^{\mathrm{l}}$ and a consistent $K^{\mathrm{u}}$-$\epsilon^{\mathrm{u}}$-separable transformation set $\Pi^{\mathrm{u}}$ for $X^{\mathrm{u}}$;

(D1) $\cap_{i=1}^n \Pi_i^{\mathrm{l}} \cap \Pi^{\mathrm{u}} \neq \emptyset$;

(E1) There exists $f^u \in \mathcal{F}^u$ such that $\{X^u, f^u\}$ is also drawn from the task distribution $\mathcal{P}(\mathcal{T}^*)$.

(A1) ensures that known and novel classes are disjoint. (B1) implies that it is meaningful to separate observations from $X_i^{\mathrm{l}}$ and $X^{\mathrm{u}}$. (C1) means that we can find good high-level features for $X_i^{\mathrm{l}}$ or $X^{\mathrm{u}}$. Based on these features, it is much easier to separate $X_i^{\mathrm{l}}$ or $X^{\mathrm{u}}$. (D1) says that the high-level features of $X_i^{\mathrm{l}}$ and $X^{\mathrm{u}}$ are shared, as demonstrated in the introduction. (E1) represents that our target task $\mathcal{T}_t = \{X^u, f^u\}$ and sampled tasks $\{\mathcal{T}_i\}_{i=1}^n$ are from the same task distribution $\mathcal{P}(\mathcal{T}^*)$. Then, we can define NCDL formally.

**Problem 2** (NCDL). *Given $\{\mathcal{T}_i = \{X_i^{\mathrm{l}}, f_i^l\}\}_{i=1}^n$ and $X^u$ defined above and assume (A1)-(E1) hold, let meta-samples $\boldsymbol{S}^{\mathrm{l}} = \{S_i^{\mathrm{l,tr}} \cup S_i^{\mathrm{l,ts}}\}_{i=1}^n$ are drawn from the $\{X_i^{\mathrm{l}}\}_{i=1}^n$, where $S_i^{\mathrm{l,tr}} \sim (X_i^{\mathrm{l}})^m$ and $S_i^{\mathrm{l,ts}} \sim (X_i^{\mathrm{l}})^k$ are the training set and the test set of the task $\mathcal{T}_i$ with the sizes $m$ and $k$, respectively, and each task $\mathcal{T}_i$ can output an inner-task clustering algorithm $\boldsymbol{A}(\boldsymbol{S}^{\mathrm{l}}) : \mathcal{X}^m \to \Pi$. In NCDL, we aim to propose a meta-algorithm $\boldsymbol{A}$ to train an inner-task clustering algorithm $\boldsymbol{A}(\boldsymbol{S}^{\mathrm{l}})$ with $\boldsymbol{S}^{\mathrm{l}}$ via minimizing $\mathcal{R}(\boldsymbol{A}(\boldsymbol{S}^{\mathrm{l}}), \{S_i^{\mathrm{l,tr}}, f_i^l\}_{i=1}^n) = \sum_{i=1}^n \tau(\boldsymbol{A}(\boldsymbol{S}^{\mathrm{l}})(S_i^{\mathrm{l,tr}})(X_i^{\mathrm{l}}), f_i^l(X_i^{\mathrm{l}}))/n$. We expect that $\boldsymbol{A}(\boldsymbol{S}^{\mathrm{l}})(S_i^{\mathrm{u}})(X^{\mathrm{u}})$ is $K^{\mathrm{u}}$-$\epsilon^{\mathrm{u}}$-separable, where $S^{\mathrm{u}}$ are observations of $X^{\mathrm{u}}$ with size $m$.*

**Remark 3.** Compared to meta-learning, NCDL aims to train an inner-task clustering algorithm $\boldsymbol{A}(\boldsymbol{S}^{\mathrm{l}}) : \mathcal{X}^m \to \Pi$ rather than a classification algorithm often used in meta learning. Besides, in NCDL, we can only observe features from the target task, while we can observe the labeled data from the new task in the meta-learning. In NCDL, $m$ is a very small number.

Then, we show that the risk used in Problem 2 can be estimated under certain conditions. Based on Problem 2, we turn the objective function $\mathcal{R}(\boldsymbol{A}(\boldsymbol{S}^{\mathrm{l}}), \{S_i^{\mathrm{l,tr}}\}_{i=1}^n)$ into a more general meta-learning risk:

$$\mathcal{R}(\boldsymbol{A}(\boldsymbol{S}^{\mathrm{l}}), P(\mathcal{T}^*)) = \mathbb{E}_{\mathcal{T}=(X,f) \sim P(\mathcal{T}^*)} \mathbb{E}_{S \sim (\mathbb{P}_X)^m} \mathbb{E}_{x \sim \mathbb{P}_X} \ell(\boldsymbol{A}(\boldsymbol{S}^{\mathrm{l}})(S)(x), f(x)), \qquad (16)$$

$\mathcal{R}(\boldsymbol{A}(\boldsymbol{S}^{\mathrm{l}}), P(\mathcal{T}^*))$ is the expectation of the generalized error w.r.t. the task distribution $P(\mathcal{T}^*)$ and can measure the performance of each inner-task clustering algorithm. In practice, the meta-clustering algorithm of NCDL is optimized by minimizing the average of the empirical error on the training tasks, called the *empirical multi-task error*:

$$\hat{\mathcal{R}}(\boldsymbol{A}(\boldsymbol{S}^{\mathrm{l}}), \{\boldsymbol{S}^{\mathrm{l}}, \mathcal{F}^{\mathrm{l}}\}) = \frac{1}{n} \sum_{i=1}^n \frac{1}{k} \sum_{x_{ij} \in S_i^{\mathrm{l,ts}}} \ell(\boldsymbol{A}(\boldsymbol{S}^{\mathrm{l}})(S_i^{\mathrm{l,tr}})(x_{ij}), f_i^{\mathrm{l}}(x_{ij})), \qquad (17)$$

where and $S_i^{\mathrm{l}} = S_i^{\mathrm{l,tr}} \cup S_i^{\mathrm{l,ts}} \sim (\mathbb{P}_{X_i^{\mathrm{l}}})^m$. Then, the generalization bound of inner-task clustering algorithm $\boldsymbol{A}(\boldsymbol{S}^{\mathrm{l}})$ of meta-based NCDL algorithms can be obtained from the uniform stability $\beta$ of the meta-algorithm $\boldsymbol{A}$.

**Definition 3** (Uniform Stability (Maurer, 2005)). *A meta-algorithm $\boldsymbol{A}$ has uniform stability $\beta$ w.r.t. the loss function $\ell$ if the following holds for any meta-samples $\boldsymbol{S}$ and $\forall i \in \{1, \ldots, n\}$, $\forall \mathcal{T} = \{X, f\} \sim \tilde{P}(\mathcal{T})$, $\forall S^{\mathrm{tr}} \sim \mathbb{P}_X^m$, $\forall S^{\mathrm{ts}} \sim \mathbb{P}_X^k$:*

$$|\hat{L}(\boldsymbol{A}(\boldsymbol{S})(S^{\mathrm{tr}})(S^{\mathrm{ts}}), f(S^{\mathrm{ts}})) - \hat{L}(\boldsymbol{A}(\boldsymbol{S}^{\setminus i})(S^{\mathrm{tr}})(S^{\mathrm{ts}}), f(S^{\mathrm{ts}}))| \le \beta,$$

*where*

$$\hat{L}(\boldsymbol{A}(\boldsymbol{S})(S^{\mathrm{tr}})(S^{\mathrm{ts}}), f(S^{\mathrm{ts}})) = \frac{1}{k} \sum_{x_j \in S^{\mathrm{ts}}} \ell(\boldsymbol{A}(\boldsymbol{S})(S^{\mathrm{tr}})(x_j), f(x_j)).$$

Given training meta-samples $\boldsymbol{S} = \{S_i^{\mathrm{tr}} \cup S_i^{\mathrm{ts}}\}_{i=1}^n$, we modify $\boldsymbol{S}$ by replacing the $i$-th element to obtain $\boldsymbol{S}^i = \{S_1^{\mathrm{tr}} \cup S_1^{\mathrm{ts}}, \ldots, S_{i-1}^{\mathrm{tr}} \cup S_{i-1}^{\mathrm{ts}}, S_i^{\mathrm{tr}'} \cup S_i^{\mathrm{ts}'}, S_{i+1}^{\mathrm{tr}} \cup S_{i+1}^{\mathrm{ts}}, \ldots, S_n^{\mathrm{tr}} \cup S_n^{\mathrm{ts}}\}$, where the replacement sample $S_i'$ is assumed to be drawn from $\mathcal{D}$ and is independent from $\boldsymbol{S}$. In addition, we modify $\boldsymbol{S}$ by removing the $i$-th element to obtain $\boldsymbol{S}^{\setminus i} = \{S_1^{\mathrm{tr}} \cup S_1^{\mathrm{ts}}, \ldots, S_{i-1}^{\mathrm{tr}} \cup S_{i-1}^{\mathrm{ts}}, S_{i+1}^{\mathrm{tr}} \cup S_{i+1}^{\mathrm{ts}}, \ldots, S_n^{\mathrm{tr}} \cup S_n^{\mathrm{ts}}\}$ In the same way, given a training set $S = \{z_1, \ldots, z_{i-1}, z_i, z_{i+1}, \ldots, z_n\}$, we can obtain $S^i = \{z_1, \ldots, z_{i-1}, z_i', z_{i+1}, \ldots, z_n\}$ and $S^{\setminus i} = \{z_1, \ldots, z_{i-1}, z_{i+1}, \ldots, z_n\}$.

**Lemma 1** (**McDiarmid Inequality**). *Let $S$ and $S^i$ defined as above, let $F : \mathcal{Z}^n \to \mathbb{R}$ be any measurable function for which there exits constants $c_i$ $(i = 1, \ldots, n)$ such that*

$$\sup_{S \in \mathcal{Z}^m, z_i' \in \mathcal{Z}} |F(S) - F(S^i)| \le c_i,$$

*then*

$$P_S[F(S) - \mathbb{E}_S[F(S)] \ge \epsilon] \le exp(\frac{-2\epsilon^2}{\sum_{i=1}^n c_i^2}).$$

**Theorem 3.** *For any task distribution $P(\mathcal{T}^*)$ and meta-samples $\boldsymbol{S}^{\mathrm{l}}$ with $n$ tasks, if a meta-algorithm $\boldsymbol{A}$ has uniform stability $\beta$ w.r.t. a loss function $\ell$ bounded by $M$, then the following statement holds with probability at least $1 - \delta$ for any $\delta \in (0, 1)$:*

$$\mathcal{R}(\boldsymbol{A}(\boldsymbol{S}^{\mathrm{l}}), P(\mathcal{T}^*)) \le \hat{\mathcal{R}}(\boldsymbol{A}(\boldsymbol{S}^{\mathrm{l}}), \boldsymbol{S}^{\mathrm{l}}) + \epsilon(n, \beta), \tag{18}$$

*where $\epsilon(n, \beta) = 2\beta + (4n\beta + M)\sqrt{\frac{\log(1/\delta)}{2n}}$.*

*Proof.* The proof of Theorem 3 mainly follows (Chen et al., 2020a).

Let $F(\boldsymbol{S}^{\mathrm{l}}) = \mathcal{R}(\boldsymbol{A}(\boldsymbol{S}^{\mathrm{l}}), P(\mathcal{T}^*)) - \hat{\mathcal{R}}(\boldsymbol{A}(\boldsymbol{S}^{\mathrm{l}}), \boldsymbol{S}^{\mathrm{l}})$ and $F(\boldsymbol{S}^{\mathrm{l},i}) = \mathcal{R}(\boldsymbol{A}(\boldsymbol{S}^{\mathrm{l},i}), P(\mathcal{T}^*)) - \hat{\mathcal{R}}(\boldsymbol{A}(\boldsymbol{S}^{\mathrm{l},i}), \boldsymbol{S}^{\mathrm{l},i})$. We have

$$|F(\boldsymbol{S}^{\mathrm{l}}) - F(\boldsymbol{S}^{\mathrm{l},i})| \le |\mathcal{R}(\boldsymbol{A}(\boldsymbol{S}^{\mathrm{l}}), P(\mathcal{T}^*)) - \mathcal{R}(\boldsymbol{A}(\boldsymbol{S}^{\mathrm{l},i}), P(\mathcal{T}^*))| + |\hat{\mathcal{R}}(\boldsymbol{A}(\boldsymbol{S}^{\mathrm{l}}), \boldsymbol{S}^{\mathrm{l}}) - \hat{\mathcal{R}}(\boldsymbol{A}(\boldsymbol{S}^{\mathrm{l},i}), \boldsymbol{S}^{\mathrm{l},i})|. \tag{19}$$

The first term in Eq. (19) can be written as

$$|\mathcal{R}(\boldsymbol{A}(\boldsymbol{S}^{\mathrm{l}}), P(\mathcal{T}^*)) - \mathcal{R}(\boldsymbol{A}(\boldsymbol{S}^{\mathrm{l},i}), P(\mathcal{T}^*))| \le |\mathcal{R}(\boldsymbol{A}(\boldsymbol{S}^{\mathrm{l}}), P(\mathcal{T}^*)) - \mathcal{R}(\boldsymbol{A}(\boldsymbol{S}^{\mathrm{l}\setminus i}), P(\mathcal{T}^*))|$$
$$+ |\mathcal{R}(\boldsymbol{A}(\boldsymbol{S}^{\mathrm{l},i}), P(\mathcal{T}^*)) - \mathcal{R}(\boldsymbol{A}(\boldsymbol{S}^{\mathrm{l}\setminus i}), P(\mathcal{T}^*))|.$$

We can upper bound the first term in Eq. (19) by studying the variation when a sample set $S_i^{\mathrm{l}}$ of training task $\mathcal{T}_i$ is deleted,

$$|\mathcal{R}(\boldsymbol{A}(\boldsymbol{S}^{\mathrm{l}}), P(\mathcal{T}^*)) - \mathcal{R}(\boldsymbol{A}(\boldsymbol{S}^{\mathrm{l}\setminus i}), P(\mathcal{T}^*))|$$
$$\le \mathbb{E}_{\mathcal{T}=(X,f)\sim P(\mathcal{T}^*)} \mathbb{E}_{S\sim(\mathbb{P}_X)^m} \mathbb{E}_{x\sim\mathbb{P}_X} |\ell(\boldsymbol{A}(\boldsymbol{S}^{\mathrm{l}})(S^{\mathrm{l}})(x), f(x)) - \ell(\boldsymbol{A}(\boldsymbol{S}^{\mathrm{l}\setminus i})(S^{\mathrm{l}})(x), f(x))|$$
$$\le \sup_{\mathcal{T}=(X,f)\sim P(\mathcal{T}^*), S\sim(\mathbb{P}_X)^m, x\sim\mathbb{P}_X} |\ell(\boldsymbol{A}(\boldsymbol{S}^{\mathrm{l}})(S^{\mathrm{l}})(x), f(x)) - \ell(\boldsymbol{A}(\boldsymbol{S}^{\mathrm{l}\setminus i})(S^{\mathrm{l}})(x), f(x))|$$
$$\le \beta.$$

Similarly, we have $|\mathcal{R}(\boldsymbol{A}(\boldsymbol{S}^{l,i}), P(\mathcal{T}^*)) - \mathcal{R}(\boldsymbol{A}(\boldsymbol{S}^{l\backslash i}), P(\mathcal{T}^*))| \leq \beta$. So the first term of Eq. (19) is upper bounded by $2\beta$. The second factor in Eq. (19) can be guaranteed likewise as follows,

$$|\hat{\mathcal{R}}(\boldsymbol{A}(\boldsymbol{S}^l), \{\boldsymbol{S}^l, \mathcal{F}^l\}) - \hat{\mathcal{R}}(\boldsymbol{A}(\boldsymbol{S}^{l,i}), \{\boldsymbol{S}^l, \mathcal{F}^l\})|$$

$$\leq \frac{1}{n}\sum_{q\neq i}\left|\frac{1}{k}\sum_{x_{qj}\in S_q^{l,ts}}(\ell(\boldsymbol{A}(\boldsymbol{S}^l)(S_q^{l,tr})(x_{qj}), f_q^l(x_{qj})) - \ell(\boldsymbol{A}(\boldsymbol{S}^{l,i})(S_q^{l,tr})(x_{qj}), f_q^l(x_{qj})))\right|$$

$$+ \frac{1}{nk}\left|\sum_{x_{ij}\in S_i^{l,ts}}\ell(\boldsymbol{A}(\boldsymbol{S}^l)(S_i^{l,tr})(x_{ij}), f_q^l(x_{ij})) - \sum_{x_{ij}\in S_i'^{,l,ts}}\ell(\boldsymbol{A}(\boldsymbol{S}^{l,i})(S_i'^{,l,tr})(x_{ij}), f_q^l(x_{ij}))\right|$$

$$\leq 2\beta + \frac{M}{n}.$$

Hence, $|F(\boldsymbol{S}^l) - F(\boldsymbol{S}^{l,i})|$ satisfies the condition of Lemma 1 with $c_i = 4\beta + \frac{M}{n}$. It remains to bound $\mathbb{E}_{\boldsymbol{S}^l}[F(\boldsymbol{S}^l)] = \mathbb{E}_{\boldsymbol{S}^l}[\mathcal{R}(\boldsymbol{A}(\boldsymbol{S}^l), P(\mathcal{T}^*))] - \mathbb{E}_{\boldsymbol{S}^l}[\hat{\mathcal{R}}(\boldsymbol{A}(\boldsymbol{S}^l), \{\boldsymbol{S}^l, \mathcal{F}^l\})]$. The first term can be written as follows,

$$\mathbb{E}_{\boldsymbol{S}^l}[\mathcal{R}(\boldsymbol{A}(\boldsymbol{S}^l), P(\mathcal{T}^*))] = \mathbb{E}_{\boldsymbol{S}^l, S_i'^{,l,tr}, S_i'^{,l,ts}}\frac{1}{k}\sum_{x_{ij}\in S_i'^{,l,ts}}\ell(\boldsymbol{A}(\boldsymbol{S}^l)(S_i'^{,l,tr})(x_{ij}), f_i^l(x_{ij})).$$

Similarly, the second term is,

$$\mathbb{E}_{\boldsymbol{S}^l}[\hat{\mathcal{R}}(\boldsymbol{A}(\boldsymbol{S}^l), \{\boldsymbol{S}^l, \mathcal{F}^l\})] = \mathbb{E}_{\boldsymbol{S}^l}\left[\frac{1}{n}\sum_{i=1}^n\frac{1}{k}\sum_{x_{ij}\in S_i^{l,ts}}\ell(\boldsymbol{A}(\boldsymbol{S}^l)(S_i^{l,tr})(x_{ij}), f_i^l(x_{ij}))\right]$$

$$= \mathbb{E}_{\boldsymbol{S}^l, S_i'^{,l,tr}}\left[\frac{1}{k}\sum_{x_{ij}\in S_i^{l,ts}}\ell(\boldsymbol{A}(\boldsymbol{S}^l)(S_i'^{,l,tr})(x_{ij}), f_i^l(x_{ij}))\right]$$

$$= \mathbb{E}_{\boldsymbol{S}^l, S_i'^{,l,tr}, S_i'^{,l,ts}}\left[\frac{1}{k}\sum_{x_{ij}\in S_i'^{,l,ts}}\ell(\boldsymbol{A}(\boldsymbol{S}^{l,i})(S_i'^{,l,tr})(x_{ij}), f_i^l(x_{ij}))\right],$$

where $\mathcal{F}^l = \{f_i^l\}_{i=1}^n$. Hence, $\mathbb{E}_{\boldsymbol{S}^l}[F(\boldsymbol{S}^l)]$ is upper bounded by $2\beta$,

$$\mathbb{E}_{\boldsymbol{S}^l}[\mathcal{R}(\boldsymbol{A}(\boldsymbol{S}^l), P(\mathcal{T}^*))] - \mathbb{E}_{\boldsymbol{S}^l}[\hat{\mathcal{R}}(\boldsymbol{A}(\boldsymbol{S}^l), \{\boldsymbol{S}^l, \mathcal{F}^l\})]$$

$$= \mathbb{E}_{\boldsymbol{S}^l, S_i'^{,l,tr}, S_i'^{,l,ts}}\left[\frac{1}{k}\sum_{x_{ij}\in S_i'^{,l,ts}}\ell(\boldsymbol{A}(\boldsymbol{S}^l)(S_i'^{,l,tr})(x_{ij}), f_i^l(x_{ij})) - \frac{1}{k}\sum_{x_{ij}\in S_i'^{,l,ts}}\ell(\boldsymbol{A}(\boldsymbol{S}^{l,i})(S_i'^{,l,tr})(x_{ij}), f_i^l(x_{ij}))\right]$$

$$\leq 2\beta.$$

Plugging the above inequality in Lemma 1, we obtain

$$P_{\boldsymbol{S}^l}[\mathcal{R}(\boldsymbol{A}(\boldsymbol{S}^l), P(\mathcal{T}^*)) - \hat{\mathcal{R}}(\boldsymbol{A}(\boldsymbol{S}^l), \boldsymbol{S}^l) \geq 2\beta + \epsilon] \leq \exp\left(\frac{-2\epsilon^2}{\sum_{i=1}^n(4\beta + \frac{M}{n})^2}\right).$$

Finally, setting the right side of the above inequality to $\delta$, the following result holds with probability of $1 - \delta$,

$$\mathcal{R}(\boldsymbol{A}(\boldsymbol{S}^l), P(\mathcal{T}^*)) \leq \hat{\mathcal{R}}(\boldsymbol{A}(\boldsymbol{S}^l), \boldsymbol{S}^l) + 2\beta + (4n\beta + M)\sqrt{\frac{\log(1/\delta)}{2n}}.$$

$\square$

By Theorem 3, the generalization bound depends on the number of the training tasks $n$ and the uniform stability parameter $\beta$. If $\beta < O(1/\sqrt{n})$, we have $\epsilon(n, \beta) \to 0$ as $n \to \infty$. Hence, given a sufficiently small $\beta$, the error $\mathcal{R}(\boldsymbol{A}(\boldsymbol{S}^l), P(\mathcal{T}^*))$ converges to training error $\hat{\mathcal{R}}(\boldsymbol{A}(\boldsymbol{S}^l), \boldsymbol{S}^l)$ as the number of training tasks $n$ grows. Theorem 3 indicates that we can minimize the risk in the NCDL problem in probability if we can control the uniform stability of a meta-algorithm (like MAML did via support-query learning (Chen et al., 2020a)) and sample the assumed tasks for training (sampler matters in meta discovery).

Table 2: Results of K-means on all four datasets.

| Dataset | CIFAR-10 (5-way) | SVHN (5-way) | CIFAR-100 (20-way) | OmniGlot (20-way) |
|---|---|---|---|---|
| 1-observation | 30.2±3.60 | 23.5±0.66 | 9.7±1.18 | 2.0±0.16 |
| 5-observation | 32.8±2.13 | 23.7±0.35 | 12.4±1.15 | 2.8±0.13 |

## D    DATASET INTRODUCTIONS AND SPLITS

CIFAR-10 dataset contains $60,000$ images with sizes of $32 \times 32$. Following (Han et al., 2019), for NCDL, we select the first five classes (i.e. airplane, automobile, bird, cat, and deer) as known classes and the rest of classes as novel classes. The amount of data from each novel class is no more than $5$. CIFAR-100 dataset contains 100 classes. Following (Han et al., 2020b), we select the first 80 classes as known classes and select the last 20 classes as novel classes.

SVHN contains $73,257$ training data and $26,032$ test data with labels 0-9. Following (Han et al., 2019), we select the first five classes (0-4) as known classes and select the (5-9) as novel classes. OmniGlot contains $1,632$ handwritten characters from 50 different alphabets. Following (Hsu et al., 2019), we select all the 30 alphabets in *background* set (964 classes) as known classes and select each of the 20 alphabets in *evaluation* set (659 classes) as novel classes.

## E    IMPLEMENTATION DETAILS

We implement all methods by PyTorch 1.7.1 and Python 3.7.6, and conduct all the experiments on two NVIDIA RTX 3090 GPUs.

**CATA.**    We use ResNet-18 (He et al., 2016) as the feature extractor and use three fully-connected layers with softmax layer as the classifier. We also use BN layer (Ioffe & Szegedy, 2015) and Dropout (Srivastava et al., 2014) in network layers. In this paper, we select the number of views $M = 3$ for all four datasets. In other words, there are three classifiers following by the feature extractor. Both the feature extractor and 3 classifier use Adam (Kingma & Ba, 2015) as their optimizer. The number of training steps is 50 and the learning rates of feature extractor and classifiers are 0.01 and 0.001 respectively. We use the tradeoff $\lambda$ of $1/3$.

**MM for NCDL.**    We use VGG-16 (Simonyan & Zisserman, 2015) as the feature extractor for all four datasets. We use SGD as meta-optimizer and general gradient descent as inner-optimizer for all four datasets. For all experiments, we sample 1000 training tasks by CATA for meta training and finetune the meta-algorithm after every 200 episodes with data of novel classes. We note that the inner-tasks of OmniGlot are sampled by in order, instead of randomly sampling like other three datasets. Thus the errors of OmniGlot only come from the training procedure, while the errors of other datasets come from both sampling procedure and training procedure. The output dimension of feature extractor $\pi_{mm}$ is set to $d_r = 512$. The meta learning rate and inner learning are 0.4 and 0.001 respectively. We use a meta batch size (the amount of training tasks per training step) of 16\8 for {CIFAR-10,SVHN}\{CIFAR-100,Omniglot}. In addition, we choose $k$ to be 10 which is suitable for all datasets. For each training task, we update the corresponding inner-algorithm by 10 steps.

**MP for NCDL.**    We use a neural network of four convolutional blocks as the feature extractor for all datasets following (Snell et al., 2017). Each block comprises a 64-filter $3 \times 3$ convolution, BN layer (Ioffe & Szegedy, 2015), a ReLU function and a $2 \times 2$ max-pooling layer. We use the same feature extractor for embedding both training data and test data and its output dimension is set to $d_r = 512$. For all experiments, we train the models via Adam (Kingma & Ba, 2015), and we use an initial learning rate of 0.001 and cut the learning rate in half every 20 steps. We train the feature extractor for 200 steps with 1000 training tasks sampled by CATA. The difference in sampling procedure and error source are the same with MM for NCDL.

## F    RESULTS OF K-MEANS

This section shows the results of our methods and all the baselines in Table 2.

Table 3: Results of NCD with abundant novel class data. In this table, we report the ACC (%)±standard deviation of ACC (%) of baselines and our methods (MM and MP) given abundant novel class data. We still evaluate these methods on four benchmarks (SVHN, CIFAR-10, CIFAR-100, and OmniGlot).

| Methods | K-means | KCL | MCL | DTC | RS | MM | MP |
|---|---|---|---|---|---|---|---|
| SVHN | 42.6±0.0 | 21.4±0.6 | 38.6±10.8 | 60.9±1.6 | 95.2±0.2 | 93.1±2.1 | 77.1±0.8 |
| CIFAR-10 | 65.5±0.0 | 66.5±3.9 | 64.2±0.1 | 87.5±0.3 | 91.7±0.9 | 92.3±0.9 | 73.2±1.9 |
| CIFAR-100 | 56.6±1.6 | 14.3±1.3 | 21.3±3.4 | 56.7±1.2 | 75.2±4.2 | 69.8±1.3 | 58.3±2.2 |
| OmniGlot | 77.2 | 82.4 | 83.3 | 89.0 | 89.1 | 88.6±0.7 | 98.4±0.2 |

## G    RESULTS OF NCD

In this section, we show the results of NCD with abundant novel-class data in Table 3. Table 3 shows that MM is comparable with the representative methods but cannot outperform the RS and MP performs worse than MM. Compared with RS, MM samples many inner-tasks for training, while RS uses the whole data. Incomplete data makes MM unable to learn the global distribution of novel classes. MP is not as well as MM on NCD tasks. As the absence of labels of novel class data, we cannot finetune the model used for calculating data embedding, which is trained by known-class data. Although this model cannot adapt to novel classes, we can calculate more accurately prototypes with *abundant* novel-class data. Hence, with MP, the results of NCD are obviously better than the results of NCDL.

## H    COMPLEXITY ANALYSIS

We give a brief analysis of time complexity for each algorithm. As MM and MP are two-step methods, we first analyze the sampling algorithm CATA, and then analyze the main parts of MM and MP.

**CATA**    The time complexity of CATA is $O(E * D/B * T)$, where $F$ is number of training tasks, $E$ is number of epochs, $D$ is size of dataset, $B$ is meta batch size, and T is the time complexity of each iteration. We can future decompose $O(T) = O(L * n)$, where $L$ is the average time complexity of each layer, and $n$ is number of layers. Then, we can decompose $O(L) = O(M * N * K^2 * H * W)$, where $M$ and $N$ are numbers of channels of input and output, $K$ is size of convolutional kernel, and $H$ and $W$ are height and weight of feature space.

**MM (Main part)**    The time complexity of MM is $O(F * E * D/B * T)$, where $F$ is number of training tasks, $E$ is number of epochs, $D$ is size of dataset, $B$ is meta batch size, and T is the time complexity of each iteration. We can future decompose $O(T) = O(L * n)$, where $L$ is the average time complexity of each layer, and $n$ is number of layers. Then, we can decompose $O(L) = O(M * N * K^2 * H * W)$, where $M$ and $N$ are numbers of channels of input and output, $K$ is size of convolutional kernel, and $H$ and $W$ are height and weight of feature space.

**MP (Main part)**    The time complexity of MP is $O(F * E * D/B * T)$, where $F$ is number of training tasks, $E$ is number of epochs, $D$ is size of dataset, $B$ is meta batch size, and T is the time complexity of each iteration. We can future decompose $O(T) = O(L * n)$, where $L$ is the average time complexity of each layer, and $n$ is number of layers. Then, we can decompose $O(L) = O(M * N * K^2 * H * W)$, where $M$ and $N$ are numbers of channels of input and output, $K$ is size of convolutional kernel, and $H$ and $W$ are height and weight of feature space.

