# OpenReview forum: "Meta Discovery: Learning to Discover Novel Classes given Very Limited Data"
_ICLR.cc/2022/Conference — ICLR 2022 Spotlight_

### Official Review · Reviewer_ueqR · 2021-10-25

**Correctness:** 4
**Technical Novelty And Significance:** 3
**Empirical Novelty And Significance:** 3
**Recommendation:** 8
**Confidence:** 3

**Main Review:**

Overall, I found this paper very interesting. The authors provide a definition and insight into a problem that is often approach with little rigor or formalization. Further, the formalization gives good motivation for the proposed methods MM and MP. However, CATA is less well explained.

My understanding is that CATA learns a low-dimension projection and set of orthogonal classifiers. It then assigns each training point to a group defined by which of the orthogonal classifiers was most strongly activated by the observation. Meta learning inner tasks are then sampled from each group. It is unclear to me what exactly this aims to accomplish. I think the authors should explain the task-sampling problem in meta learning more completely.

Typo on page 6. "For $z_i = \pi_{mm}(x_i)$ and $z_i = \pi_{mm}(x_j)$" should be $z_i = \pi_{mm}(x_i)$ and $z_j = \pi_{mm}(x_j)$".



**Summary Of The Paper:**

The authors provide a formal definition of the L2DNC task, give proofs of useful insight into the problem, and empirically demonstrate novel methods. The authors use several baselines and ablation to demonstrate the value of the proposed methods. The theoretical insights suggest that common high level features are necessary to solve the L2DNC task, which motivates the proposed approaches based on low dimensional non-linear projection of the data. Similarly the clustering-centric definition of the problem motivates the similarity learning methods presented.

**Summary Of The Review:**

The theoretical framing of the L2DNC problem is valuable, as it can help us understand what is necessary to solve this problem effectively.

---

> ### Author Response · Authors · 2021-11-20
> **Response to Reviewer ueqR**
>
> Thanks for your constructive comments! We have addressed your concerns in the following.
>
> >Q1. However, CATA is less well explained. My understanding is that CATA learns a low-dimension projection and set of orthogonal classifiers. It then assigns each training point to a group defined by which of the orthogonal classifiers was most strongly activated by the observation. Meta learning inner tasks are then sampled from each group. It is unclear to me what exactly this aims to accomplish. I think the authors should explain the task-sampling problem in meta learning more completely.
>
> A1. You have understood the process of CATA correctly. We will merge your understanding into our paper. We will explain why CATA can effectively boost L2DNC/L2DNCL. CATA maintains a certain amount of orthogonal (i.e., independent) classifiers, and they have different decision boundaries, indicating that they have different views. For a data point $(x,y)$, the largest $P_i(y|x)$ means that the dominant view of $x$ is the $i^{th}$ view. As we know, L2DNC/L2DNCL is a clustering problem. In Figure 3, we claim that each data point has its dominant clustering rule. For L2DNCL, each inner task has only very few data (i.e., N*K). If these data have various dominant rules, they will interfere with each other, making the inner algorithm unable to learn a common rule. If we sample data with CATA, the data in each inner task have the same dominant clustering rule and can be clustered more precisely.
>
> >Q2. Typo on page 6.
>
> A2. Thanks for your carefulness. We will modify this typo in the modified version and check our paper again.

---

> > ### Comment · Reviewer_ueqR · 2021-11-24
> > **Thank you for the clarification**
> >
> > I have read the other reviews and author responses. I am satisfied that the authors have address my questions and the intuitive explanation of the method has been improved.

---

### Official Review · Reviewer_iZHD · 2021-11-02

**Correctness:** 4
**Technical Novelty And Significance:** 4
**Empirical Novelty And Significance:** 4
**Recommendation:** 8
**Confidence:** 5

**Main Review:**

Pros:

+ This paper is well-written and easy to follow. The motivation of the new setting of NCD is clear.

+ This paper considers the problem of NCD in a new perspective and rigorously formulates the NCD setting.

+ A theoretical analysis is provided to demonstrate when the NCD setting can be solved. In addition, they reveal that NCD can be solved if known and novel classes are related. They also show a result to see how NCD fails if one key assumption is not satisfied. The authors give detailed algorithms and the source code, which can help readers better understand and reproduce the proposed method.

+ Extensive experiments are provided to verify the effectiveness of the proposed method. Also, all the results include the standard deviation, which is important for reflecting the real improvements on the CIFAR and SVHN datasets.

Cons:

- It is interesting to see a formulation of NCD problem. In this formulation, we can know when the labeled data can help cluster novel-class data. However, the connection between the theory and solution is not very strong. It is better to move some parts in the Appendix to the main context. Using meta-learning to define L2DNCL is also an interesting part.

- In Definition 1, why do you need to set an equation here? Is it better to consider an inequality? It is like that the separation is less and \epsilon? A similar question follows in Definition 2.

- It is not appropriate to say “Impossibility Theorem for Previous Setting”. I admit that previous studies do not consider NCD based on some basic assumptions, but they still propose many working algorithms for NCD. Thus, I recommend deleting the “Previous Setting”. You can add some explanations below Theorem 2.

- The pipeline of meta-learning should be introduced clearly. The connection between Sections 3 and 4 is somehow weak. I understand that meta-learning is suitable for few-shot situations. However, it is not good to use this as the motivation of your method. It is not easy to know how to use the output of Algorithm 2. I found that L2DNCL is demonstrated in Appendix, which is much clear. You can consider swapping some descriptions from Appendix to the main context.

- When training the pairwise loss, did you use different outputs for the labeled data and unlabeled data? That is, using outputs with C_l-dim for the labeled data while outputs with C_u-dim for the unlabeled data.

- How to sample the training data without the CATA? I did not see the trivial sampling procedure for MEDI.

- How about include the results of K-means in all figures? I did not get the point why you need Table 1.

**Summary Of The Paper:**

This paper considers the problem of novel class discovery (NCD). Different from the original setting of NCD, this paper reconsiders the assumptions behind NCD and defines a new yet more practical setting, which can significantly reduce the number of unlabeled data needed for training novel classes. In addition, this paper presents a meta-learning-based approach to address the new setting, which achieves consistent improvements on four public datasets. They also provide a theory to reveal why we need to assume that known and novel classes should share semantic features.

**Summary Of The Review:**

Overall, this is an interesting paper, which reformates the problem of NCD to a more realistic setting. In addition, this paper builds a connection between meta-learning and NCD and presents an effective approach to address the introduced problem. Extensive experiments are provided to verify the effectiveness of the proposed method. Some statements and implementation details are not very clear, I would like to see the response of the authors.

---

> ### Author Response · Authors · 2021-11-20
> **Response to Reviewer iZHD**
>
> Thanks for your constructive comments. We have addressed your concerns in the following.
>
> >Q1. However, the connection between the theory and solution is not very strong. It is better to move some parts in the Appendix to the main context.
>
> A1. Thanks for your great advice. The connection between theory and solution mainly lies in the assumption (D). Assumption (D) indicates that the high-level features of $X^l$ and $X^u$ are shared, which is the same with meta-learning, so as to yield our solution. We will strengthen the connection between Section 3 and Section 4.
>
> >Q2. In Definition 1, why do you need to set an equation here? Is it better to consider an inequality? It is like that the separation is less and $\epsilon$? A similar question follows in Definition 2.
>
> A2. It is a very good question. When we first tried to define the K-epsilon separation, we adopt the inequality rather than the equality since the inequality seems more natural. Then, however, we find that the inequality mixes up too much information such that we cannot prove our theorems. More importantly, the equality actually gives a level set for each epsilon and can demonstrate the separation situation more precisely than the inequality. Hence, we choose the equality in the end.
>
> >Q3. It is not appropriate to say “Impossibility Theorem for Previous Setting”. You can add some explanations below Theorem 2.
>
> A3. Thanks for your great advice. The name of Theorem 2 is indeed not proper. Although previous literatures did not clarify this assumption, they also did not claim that L2DNC was solvable without assumption (D). We will remove “Previous Settings” following your advice.
>
> >Q4.  The pipeline of meta-learning should be introduced clearly. The connection between Sections 3 and 4 is somehow weak. I understand that meta-learning is suitable for few-shot situations. However, it is not good to use this as the motivation of your method. It is not easy to know how to use the output of Algorithm 2. I found that L2DNCL is demonstrated in Appendix, which is much clearer. You can consider swapping some descriptions from Appendix to the main context.
>
> A4.  We will talk about the general pipeline of meta-learning for L2DNC in the modified version and strengthen the connection between Section 3 and Section 4.
>
> We define L2DNC in Section 3, and  demystify its underlying assumption. Based on this assumption, we naturally link L2DNC with meta-learning, and then we move on to Section 4. In the modified version, we will analyze their connection in detail.
>
> Algorithm 2 can obtain a meta clustering algorithm, which can quickly adapt to a new task given few data.
>
> Constrained by page limitation, we mainly discuss L2DNCL in Appendix C. In the modified version, we will try to give more discussion of L2DNCL in the main text.
>
> >Q5. When training the pairwise loss, did you use different outputs for the labeled data and unlabeled data? That is, using outputs with C_l-dim for the labeled data while outputs with C_u-dim for the unlabeled data.
>
> A5. The inner tasks of both MM and MP are “N-way K-obsv” clustering problems, thus the output dimension of our clustering algorithms are N-dim.
>
> >Q6. How to sample the training data without the CATA? I did not see the trivial sampling procedure for MEDI.
>
> A6. Without CATA, we will sample inner tasks from all the known-class data randomly, taking no account of their dominant clustering rules.
>
> >Q7. How about including the results of K-means in all figures? I did not get the point why you need Table 1.
>
> A7. The accuracy of K-means are relatively low, especially for OmniGlot. We have tried to include the results of K-means in corresponding figures. However, the gap between K-means and MM and MP is too large, thus reporting them in the same figure will mislead the readers. Taking OmniGlot for an example, the result of OmniGlot in the 20-way 5-obsv task is 98.4%, but it looks like 100% if the results of K-means are included.

---

> > ### Comment · Reviewer_iZHD · 2021-11-25
> > **My concerns are well addressed.**
> >
> > Thanks for your response. My concerns are well addressed. I also have carefully read the comments from other reviewers. To this end, I would like to keep my acceptance score.

---

### Official Review · Reviewer_V9Yq · 2021-11-02

**Correctness:** 4
**Technical Novelty And Significance:** 3
**Empirical Novelty And Significance:** 4
**Recommendation:** 8
**Confidence:** 3

**Main Review:**

Learning to discover novel classes (L2DNC) is a challenging task,
where the learnability conditions are not entirely clear. The work
contributes to formalize some intuitive notions of learnability in
terms of epsilon-separation and consistent transformation sets,
listing the assumptions that are needed for L2DNC to be learnable.

The need for a shared transformation/representation, that is one of
the results of the formalization, is sensible but clearly not
novel. On the other hand, the author introduce a multi-view learning
that intuitively makes sense as a way to help clustering to focus on
the correct semantic features, but its connection with the formal
treatment is less clear.

Results are reasonably well-discussed, and include both few-data (the
setting of the paper) and large-data results (where performance
difference shrink, as expected) are well as ablation studies.

Minor points:

Why "X is K-epsilon-separation" ? should it be K-epsilon-separable or something?



**Summary Of The Paper:**

The paper introduces meta-discovery as a way to adapt meta-learning
strategies to the problem of learning to discover novel classes. It
also presents a formalization of the problem that helps shading light
on the conditions under which it is learnable.

The authors additionally introduce a novel sampling strategy aimed at
sampling data having the same "dominant" view to help the following
clustering procedure.

The experimental evaluation shows advantages over existing recent
competitors when the number of examples for unseen classes is small.

**Summary Of The Review:**

A useful formalization of the conditions under which L2DNC works.

An intuitively sensible multi-view based meta discovery algorithm
shown to outperform existing alternatives in the low-data regime.

---

> ### Author Response · Authors · 2021-11-20
> **Response to Reviewer V9Yq**
>
> Thanks for your constructive comments. We have addressed your concerns in the following.
>
> >Q1. The need for a shared transformation/representation, that is one of the results of the formalization, is sensible but clearly not novel.
>
> A1. The shared transformation/representation is indeed used in many generalization-related problems. However, one contribution of our paper is to demystify this implicit assumption of L2DNC, so as to define this problem in a meaningful and strict way and give an efficient solution.
>
> >Q2. The connection between CATA and formal treatment is less clear.
>
> A2. CATA as a sampler that is different from samplers used in meta learning. Trivial samplers in meta learning sample data randomly, taking no account of any properties of data. If the data in an inner-task have various dominant clustering rules, they will interfere with each other, leading to a low clustering accuracy. However, CATA maintains a certain number of classifiers, and they have different decision boundaries, yielding different views. We propose to treat the dominant view as the dominant clustering rule. Then, the training data are assigned to a certain number of groups, and data in one group have the same dominant clustering rule. Thus, CATA tends to sample data from one group to compose an inner-task.
>
> It is the first time to consider to collect data with the same clustering rule. Indeed, CATA is an intuitive sampler, and we are still working on finding a proper formal definition of the clustering rule. Then, we might analyze CATA theoretically. Fortunately, our ablation study has verified that CATA might help collect data with the same clustering rule, which encourages us finding the reason why CATA succeed. Thanks for your great comment. We will list this point as an important future work in our paper.
>
> >Q3. Why "X is K-epsilon-separation"? should it be K-epsilon-separable or something?
>
> A3. This is great advice. “K-epsilon-separation” is a property for one random variable. Thus, naming it by “K-epsilon-separable” as you suggested will be more appropriate.

---

### Official Review · Reviewer_o2Tq · 2021-11-04

**Correctness:** 4
**Technical Novelty And Significance:** 4
**Empirical Novelty And Significance:** 3
**Recommendation:** 8
**Confidence:** 4

**Main Review:**

Pros:

1.  The analysis of solvability is very interesting. Since we only need to cluster novel classes, it seems that introducing additional data is useless. In this paper, the authors answer this question and point when L2DNC is a meaningless problem. This point might affect the development of the L2DNC. Confirming the boundary of solvability of L2DNC is important since we can naturally do the clustering task with novel classes themselves.

2. Considering few-observation situations make L2DNC methods be used in more practical scenarios.

3. The proposed method can improve the performance with large margins, which is important to the field. CATA is also a new sampler to the meta-learning field (ablation studies verify the effectiveness of CATA as well).

4. Interacting with meta-learning provides more possibilities to improve the performance according to SOTA meta-learning methods. The meta discovery is a general framework to address L2DNCL.

Cons:

1. I feel struggling when I first read this paper. Figure 1(a) misleads me and lets me think that L2DNC aims to detect novel classes. You should explain L2DNC at first in the caption of Figure 1.

2. I cannot follow the benefits of the sampling process (Y->X). The difference between X->Y and Y->X should be discussed in detail in introduction (it is important to make readers understand them in the introduction).

3. K-epsilon-separation r.v. is a very interesting concept. However, I am not sure why we need \tau=\epsilon rather than \tau<\epsilon. The latter seems more natural.

4. There is a typo in (2), it should be \mathbb{P}_{\pi(X)} rather than \mathbb{P}_{X}, which makes me very confused when I read (2).

5. From two theorems, Assumption (D) is indeed an important assumption for L2DNC problem. However, the connection between (D) and Y->X is not very clear. I find that (D) is not the sufficient and necessary condition of Y->X. More discussions are needed here. You cannot simply say that Y->X means (D). From my experience, (D) contains more situations. So, what are these situations?

6. The condition used in Theorem 2 should be explained in natural language.

7. Directly introducing meta-learning (using “few”) is not good. At least, you should introduce meta-learning based on (D) or Y->X.

8. Expect for CATA, are there new components in MM or MP?

9. Experiments look nice, but I am interesting to know why MP can have such good performance in OmniGolt.


**Summary Of The Paper:**

Learning to discover novel class is a very challenging task and a new research topic in recent years. In this task, a known-class dataset can be used to help cluster novel classes. “Novel” means that there are no overlaps between novel classes and known classes. This setting looks ill-defined since it is not clear why we need known classes to help cluster novel classes. However, this paper answers this question based on a novel concept: K-epsilon separation and points out when this setting is ill-defined/well-defined. Based on this contribution, this paper is above the acceptance borderline.

Nevertheless, the presentation should be polished. Although I understand the setting and the contribution in the end, the presentation flow is not smooth. Some typos make me struggle when I read this paper. Besides, the connection between sampling process and Assumption (D) should be discussed deeply. I find that they might not the same thing.


**Summary Of The Review:**

The current version is a borderline paper because some important parts seem unclear to me. The presentation should be polished, and the connections among sections should be strengthened. I expect a better version after the discussion period.

---

> ### Author Response · Authors · 2021-11-20
> **Response to Reviewer o2Tq (Part 2)**
>
> >Q6. The condition used in Theorem 2 should be explained in natural language.
>
> A6. We have explained assumption (A)~(D) in natural language after “Problem Setup of L2DNC”, and then we will explain $\max_{{\pi}\in\Pi^l}\tau({\pi}(X^u),f^u(X^u))<\min_{{\pi}\in\Pi-\Pi^l}\tau({\pi}(X^l),f^l(X^l))$ (1) in detail. (1) indicates that the worst case of clustering novel classes with transformations that are suitable for known classes is better than the best case of clustering known classes with transformations that are not suitable for known classes. In addition, $\epsilon^l\le\epsilon^u$ indicates that transformations in $\Pi^l$ have better performance than transformations in $\Pi^u$.
>
> >Q7. Directly introducing meta-learning (using “few”) is not good. At least, you should introduce meta-learning based on (D) or Y->X.
>
> A7. This is very good advice. In the last of Section 3, we have clarified that meta-learning also assumes that known and novel classes share the high-level semantic features. This is exactly the same with assumption (D). Thus, we will link them in the modified version.
>
> >Q8. Except for CATA, are there new components in MM or MP?
>
> A8. Except for CATA, there also exists differences in MM and MP. For MM, the loss function of inner-tasks is designed for clustering problems, instead of cross-entropy loss in MAML. For MP, compared with ProtoNet, MP does not have the finetuning process, i.e., using labeled novel-class data to finetune the prototypes learned from labeled known-class data.
>
> >Q9. Experiments look nice, but I am interesting to know why MP can have such good performance in OmniGolt.
>
> A9. OmniGlot is a representative and simple benchmark in the field of few-shot learning (this is the reason why we choose it as one of our benchmark datasets). In global class representations (GCR) [r1], for 20-way 1-shot, GCR achieves classification accuracy of 99.63%. In adaptive posterior learning (APL) [r2], for 20-way 5-shot, APL achieves classification accuracy of 99.9%. As a popular benchmark in the field of few-shot learning, OmniGlot has been solved when labels are available. Therefore, our results are common and reasonable.
>
> References
>
> [r1] Li, Aoxue, et al. "Few-shot learning with global class representations." ICCV. 2019.
>
> [r2] Ramalho, Tiago, and Marta Garnelo. "Adaptive posterior learning: few-shot learning with a surprise-based memory module." ICLR. 2019.

---

> > ### Comment · Reviewer_o2Tq · 2021-11-24
> > **Thanks for the clarification**
> >
> > I have read the authors' responses and other reviewers' comments. My concerns have been properly addressed. Besides, I do believe that this is an understudied topic and this paper is a valuable contribution to the field.
> >
> > Thus, I raised my score to 8.

---

> ### Author Response · Authors · 2021-11-20
> **Response to Reviewer o2Tq (Part 1)**
>
> Thanks for your constructive comments. We have addressed your concerns in the following.
>
> >Q1. I feel struggling when I first read this paper. Figure 1(a) misleads me and lets me think that L2DNC aims to detect novel classes. You should explain L2DNC at first in the caption of Figure 1.
>
> A1. Your advice is very valuable. To avoid Figure 1(a) misleading readers, we will first explain what L2DNC is in the caption, and then discuss why formulating L2DNC in a sampling way is more meaningful.
>
> >Q2. I cannot follow the benefits of the sampling process (Y$\to$X). The difference between X$\to$Y and Y$\to$X should be discussed in detail in introduction (it is important to make readers understand them in the introduction).
>
> A2. If L2DNC is labeling in causality, L2DNC may be not theoretically solvable (Theorem 2). That is to say, we have obtained many unlabeled data (to be annotated), and we have to go through them to find novel-class data. In this process, novel classes may differ from known classes very much due to the diversity of samples in the unlabeled dataset, contributing no useful information to the clustering of novel-class data. However, if L2DNC is sampling in causality, novel-class data are collected in the same way of sampling known-class data, thus they are highly related and share high-level semantic features, which makes L2DNC theoretically solvable (Theorem 1).
>
> We will follow your valuable advice to clarify the difference between X$\to$Y and Y$\to$X in the 2nd paragraph and the 3rd paragraph in the Introduction.
>
> >Q3. K-epsilon-separation r.v. is a very interesting concept. However, I am not sure why we need $\tau=\epsilon$ rather than $\tau<\epsilon$. The latter seems more natural.
>
> A3. It is a very good question. When we first tried to define the K-epsilon separation, we adopt the inequality rather than the equality since the inequality seems more natural. Then, however, we find that the inequality mixes up too much information such that we cannot prove our theorems. More importantly, the equality actually gives a level set for each epsilon and can demonstrate the separation situation more precisely than the inequality. Hence, we choose the equality in the end.
>
> >Q4. There is a typo in (2), it should be $\mathbb{P}_{\pi(X)}$ rather than $\mathbb{P}_X$, which makes me very confused when I read (2).
>
> A4. Your understanding is right. We will correct this typo and carefully check our paper again.
>
> >Q5. From two theorems, Assumption (D) is indeed an important assumption for L2DNC problem. However, the connection between (D) and Y$\to$X is not very clear. I find that (D) is not the sufficient and necessary condition of Y$\to$X. More discussions are needed here. You cannot simply say that Y$\to$X means (D). From my experience, (D) contains more situations. So, what are these situations?
>
> A5. Your understanding is right. After Theorem 2, we will discuss the relation between Assumption (D) and Y$\to$X and show other situations that Assumption (D) still holds. For example, if there are some constraints to make the unlabeled data (to be annotated) are obtained in the same scenario, then data generated by X$\to$Y also satisfy the Assumption (D).  If Assumption (D) holds, then we can basically guarantee L2DNC is solvable.

---

### Author Response · Authors · 2021-11-23
**Revision has been uploaded**

We greatly appreciate the valuable comments of all reviewers. We summerize all the major revisions in the following.

>Section 1: Introduction

1. Discuss the difference between $X\to Y$ and $Y\to X$ in the 2nd and 3rd paragraphs.

2. Explain L2DNC at first in the caption of Figure 1.

>Section 3: Assumptions behind L2DNC and Analysis of Solvability

1. Discuss more about the common assumption of meta-learning and L2DNC (i.e., Assumption (D)), making the transition from Section 3 to Section 4 smoother.

2. Discuss the relationship between Assumption (D) and $Y\to X$.

3. Change the name of Theorem 2 from "Impossibility Theorem for Previous Setting" to "Impossibility Theorem".

4. Explain the conditions of Theorem 2 intuitively.

5. Correct the typo in Eq. (2).

>Section 4: Meta Discovery for L2DNCL

1. Merge the Reviewer ueqR's understanding about CATA to our paper.

2. Claim that CATA is a heuristic method, and we will give clustering rule a formal definition and explore the reason why CATA succeed theoretically in the future.

---

### Decision · Program_Chairs · 2022-01-20

**Decision:**

Accept (Spotlight)

**Comment:**

All reviewers believe that this paper is valuable, and the authors have made a significant, careful contribution.

Some suggestions from the area chair:
- "in causality" is not a standard technical term and also not non-technical idiomatic English, so it should be explained the first time it is used.
- The authors should briefly cite and discuss research on so-called positive and unlabeled (PU) learning. This seems like the special case where there is exactly one known class and one novel class. The distinction between sampling in causality and labeling in causality appears in the PU literature, though not under this name.
- The authors could also mention the obvious but surprising point that if data are generated by two clusters, then a classifier can be learned using exactly one labeled example--not even one from each class.
- I have read the reference EJ A’Court Smith. Discovery of remains of plants and insects. _Nature_, 1874 and I fail to see its relevance. It is only one paragraph. Work from the 1800s should not be cited merely to suggest a veneer of scholarliness.
- The writing uses italics for emphasis much too often.